# The ion channel *ppk301* controls freshwater egg-laying in the mosquito *Aedes aegypti*

Benjamin J Matthews[1,2†], Meg A Younger[1,3†], Leslie B Vosshall[1,2,3*]

[1]Laboratory of Neurogenetics and Behavior, The Rockefeller University, New York, United States; [2]Howard Hughes Medical Institute, New York, United States; [3]Kavli Neural Systems Institute, New York, United States

**Abstract** Female *Aedes aegypti* mosquitoes are deadly vectors of arboviral pathogens and breed in containers of freshwater associated with human habitation. Because high salinity is lethal to offspring, correctly evaluating water purity is a crucial parenting decision. We found that the DEG/ENaC channel *ppk301* and sensory neurons expressing *ppk301* control egg-laying initiation and choice in *Ae. aegypti*. Using calcium imaging, we found that *ppk301*-expressing cells show *ppk301*-dependent responses to water but, unexpectedly, also respond to salt in a *ppk301*-independent fashion. This suggests that *ppk301* is instructive for egg-laying at low-salt concentrations, but that a *ppk301*-independent pathway is responsible for inhibiting egg-laying at high-salt concentrations. Water is a key resource for insect survival and understanding how mosquitoes interact with water to control different behaviors is an opportunity to study the evolution of chemosensory systems.

DOI: https://doi.org/10.7554/eLife.43963.001

*For correspondence:
leslie@rockefeller.edu

†These authors contributed equally to this work

Competing interests: The authors declare that no competing interests exist.

## Introduction

A female *Ae. aegypti* mosquito must take a blood-meal to develop a batch of eggs. Many strains of *Ae. aegypti* are anthropophilic, meaning that they target humans as their preferred blood source (*Ponlawat and Harrington, 2005*; *McBride et al., 2014*). Once proteins and other nutrients in the blood have been converted into eggs, a female mosquito must find a suitable body of freshwater to lay these eggs (*Day, 2016*). Larval and pupal stages of *Ae. aegypti* are aquatic and thus the choice of egg-laying site is a primary determinant of offspring survival (*Christophers, 1960*; *Powell and Tabachnick, 2013*). In the field, *Ae. aegypti* often lay eggs in small containers of freshwater, such as drinking water storage containers, discarded tires, clogged gutters, and other by-products of human settlement, likely contributing to their pernicious ability to adapt and colonize human settlements across the globe (*Kraemer et al., 2015*).

The act of choosing an egg-laying site in many insects, including *Ae. aegypti*, is a deliberate and evaluative behavior. A female *Ae. aegypti* mosquito with a fully developed batch of eggs will use elevated humidity and bacterial volatiles to locate water at a distance (*Bentley and Day, 1989*; *Barbosa et al., 2010*). Once in close-range, a mosquito contacts water to evaluate its suitability for egg-laying by sensing a wide variety of cues that include salinity, food, bitter toxins, and animal-derived chemical signals indicating the density of conspecific larvae and pupae or the presence of predators (*Hudson, 1956*; *Benzon and Apperson, 1988*; *Bentley and Day, 1989*; *Zahiri and Rau, 1998*; *Pamplona et al., 2009*; *Afify and Galizia, 2015*; *Day, 2016*; *Zuharah et al., 2016*). The physiological mechanisms by which a mosquito senses the presence of water and evaluates its composition to safely nurture her offspring remain unknown.

**eLife digest** When they bite humans, mosquitoes can transmit dangerous diseases. For example, the *Aedes aegypti* mosquito spreads the viruses that cause yellow fever, Zika and dengue fever. Only the female mosquitoes feed on blood so they can obtain the nutrients they need to develop their eggs. Once they are ready, the insects lay their eggs in carefully selected sites where fresh water collects: if instead they choose places where the water is too salty, their offspring will die.

To find a suitable site, a mosquito 'tastes' the water by dipping in its legs and mouthparts, which activates the insect's sensory neurons and sends signals to its brain. However, it remains unclear exactly how the mosquitoes can distinguish between fresh and salty water. To address this question, Matthews, Younger and Vosshall used a combination of genetic and imaging approaches to study female *Ae. aegypti* mosquitoes.

These experiments identified a gene known as *ppk301* that is necessary for the mosquitoes to successfully lay their eggs in the right type of water. Mutant *Ae. aegypti* mosquitoes lacking the *ppk301* gene did not properly lay eggs in fresh water even when given the opportunity.

Further experiments found that *ppk301* was present in specific neurons in the legs and mouthparts of the mosquitoes. In leg neurons, *ppk301* played a crucial role in sensing the presence of water and in stimulating the mosquitoes to lay eggs in water containing low levels of salt. However, these cells still responded to salt, even when lacking *ppk301*: other unidentified genes must therefore also be involved in preventing the mosquitoes from breeding in water that is too salty.

Every year, *Aedes aegypti* and other mosquitoes infect hundreds of millions of people and cause 500,000 deaths. Knowing exactly how mosquitoes breed could help to develop traps, repellents and other strategies to stop the insects from multiplying.

DOI: https://doi.org/10.7554/eLife.43963.002

Here, we show that the DEG/ENaC channel *ppk301* is required for mosquitoes to exploit freshwater egg-laying substrates. When *ppk301* mutant females contact water, they do not lay eggs as readily as wild-type animals and are more likely to make aberrant decisions between freshwater and saltwater at concentrations that impair offspring survival. We developed a CRISPR-Cas9-based genetic knock-in strategy to build genetic tools for labeling and imaging molecularly defined populations of neurons and generated a reporter line expressing the QF2 transcriptional activator from the endogenous *ppk301* locus. We found that *ppk301* is expressed in sensory neurons in legs and proboscis, appendages that directly contact water, and that sensory afferents from these *ppk301*-expressing neurons project to central taste centers. Using in vivo calcium imaging with the genetically encoded calcium sensor GCaMP6s at the axonal terminals of *ppk301*-expressing neurons in the ventral nerve cord, we found that *ppk301* neurons respond to freshwater and that this response was almost entirely abolished in *ppk301* mutant animals. Surprisingly, these projections were also activated by salt in both wild-type and *ppk301* animals. We propose that *ppk301* drives egg-laying at low-salt concentrations and that egg-laying is inhibited by a *ppk301*-independent pathway at high-salt concentrations.

## Results

We observed *Ae. aegypti* egg-laying behavior in the laboratory to understand the range of sensory information that is integrated to guide the final decision to lay an egg. A female *Ae. aegypti* mosquito with a fully developed batch of eggs will first evaluate a potential egg-laying site by physical contact with some combination of sensory tissues that include all three pairs of legs and her proboscis. If a site is deemed suitable, she will lay eggs singly on the moist substrate above the waterline (*Hudson, 1956*) (*Figure 1A*, *Video 1*). When we used a mesh barrier to block access to liquid water, females did not lay eggs, even though they had free access to the moist filter paper substrate (*Figure 1B,C*). Together, this suggests that direct contact with liquid water is necessary to stimulate egg-laying in *Ae. aegypti* and we set out to probe the molecular and cellular basis of this water-sensation to better understand this critical mosquito behavior.

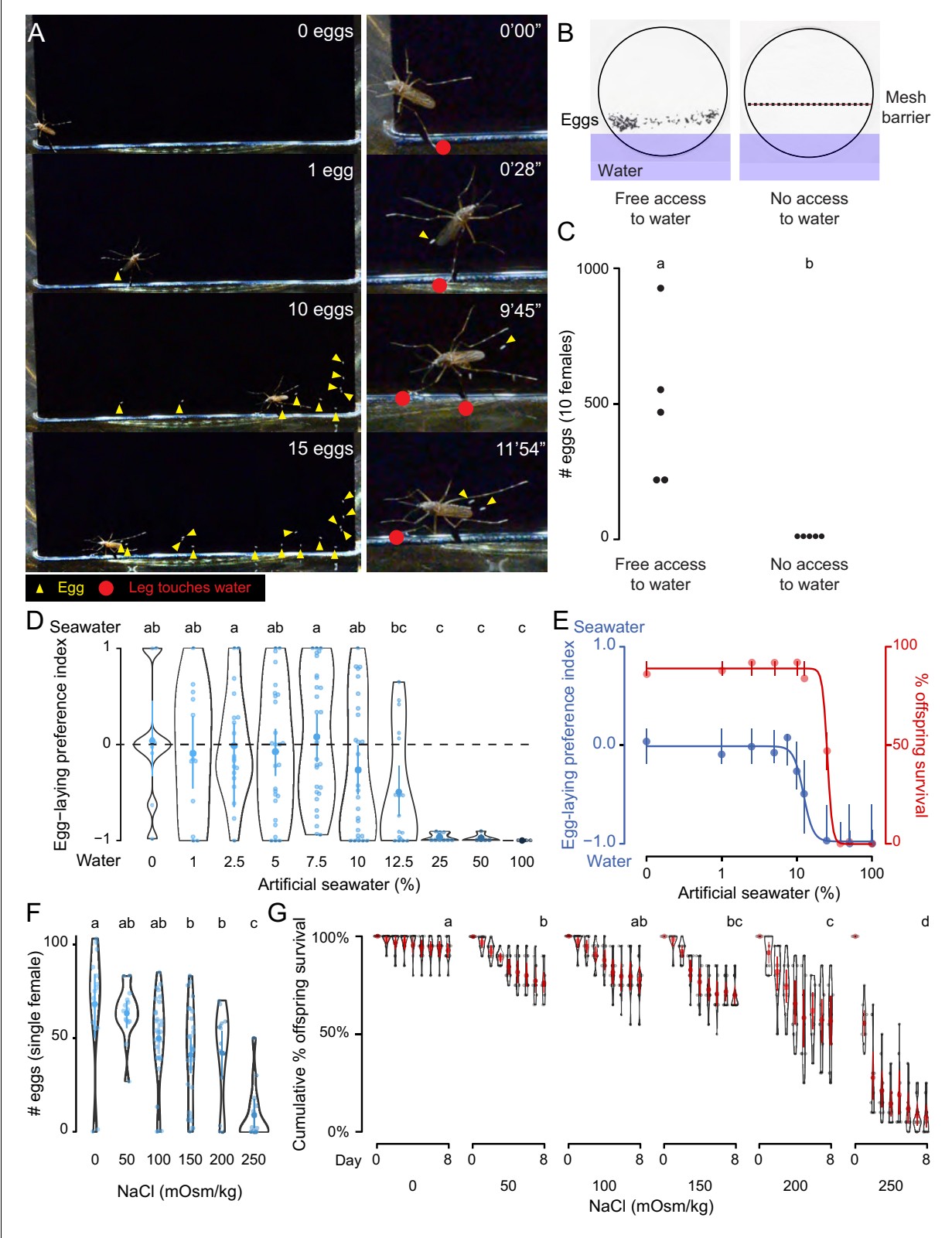

**Figure 1.** *Ae. aegypti* mosquitoes use freshwater contact to guide egg-laying, ensuring offspring survival. (**A**) Still video frames of *Ae. aegypti* female with newly laid eggs (yellow triangles) and contact between sensory appendages and liquid water (red circles) indicated. Right panels are magnification of animal in left panels. (**B**) Egg-papers (black circles) from 10 females allowed to lay eggs for 18 hr with access to water (left) or access blocked by metal mesh (right). (**C**) Eggs laid after 18 hr under conditions in b. n = 5; each dot represents data from a group of 10 females. (**D**) Egg-laying

*Figure 1 continued on next page*

*Figure 1 continued*

preference of single females between water and increasing concentrations of artificial seawater. Mean ±95% confidence interval. n = 4–29/concentration. (E) Four-parameter log-logistic curve fit to egg-laying preference in D (blue, n = 4–29/concentration) or offspring surviving (red, n = 5/concentration) in indicated concentrations of artificial seawater. Mean of the data is presented along with 95% confidence intervals of the model fit. (F) Eggs laid by single females given access to liquid of the indicated NaCl concentration. Mean ±95% confidence interval. n = 14–28/concentration. (G) Cumulative survival of larvae in indicated concentration of NaCl. Mean ±95% confidence interval. n = 9 groups/concentration. In a given panel, data labeled with different letters are significantly different from each other; p<0.05, two-way paired t-test (C), ANOVA followed by Tukey's HSD (D, F) or ANOVA followed by Tukey's HSD of survival at day 8 (G).

DOI: https://doi.org/10.7554/eLife.43963.003

As a consequence of their global spread, *Ae. aegypti* are faced with diverse habitats with a wide variety of potential egg-laying sites. For example, they can be found in abundance in a number of coastal regions rich with standing saltwater (*Ramasamy et al., 2014*; *de Brito Arduino et al., 2015*). To mimic the choice between freshwater and seawater in the lab, we developed a two-choice assay in which individual blood-fed females were placed in a container with a divided Petri dish filled with deionized water on one side and varying concentrations of a chemically defined artificial seawater solution (*Kester et al., 1967*) on the other (*Figure 1D*, *Supplementary file 1*). Mosquitoes showed no significant preference between deionized water and dilute seawater up to 10%, with individual mosquitoes either picking a solution at random or distributing their eggs between both solutions (*Figure 1D*). However, they showed a strong dose-dependent aversion to higher concentrations of seawater, with an $IC_{50}$ of 12.25% seawater (*Figure 1D,E*). Females showed near-complete aversion to 25–100% seawater (*Figure 1D,E*). These choices have consequences for the offspring. When we measured survival of offspring reared in varying concentrations of seawater, we found dose-dependent lethality ($LD_{50} = 25.23\%$) (*Figure 1E*). To simplify the stimulus, we used sodium chloride (NaCl), the predominant salt in artificial seawater, in all subsequent experiments. Females showed dose-dependent inhibition of egg-laying with increasing concentrations of NaCl (*Figure 1F*) when they were only given access to a single concentration, suggesting that the preference for freshwater may be driven in part by an aversion to laying eggs in saltwater. Similar to artificial seawater, NaCl produced a dose-dependent decrease in offspring survival (*Figure 1G*). This demonstrates that the female mosquito's choice of freshwater or saltwater correlates with the survival of her offspring, making this an essential decision for the propagation and fitness of the species.

In a search for genes that control *Ae. aegypti* freshwater egg-laying, we reasoned that animals carrying a mutation in an egg-laying preference gene would fail to lay eggs in freshwater, inappropriately lay in saltwater, or both. *pickpocket* (*ppk*) genes are members of the Degenerin/ENaC channel superfamily and encode cation channel subunits that function as putative mechanoreceptors and chemoreceptors for a wide array of stimuli including pheromones, liquid osmolality, and salt (*Adams et al., 1998*; *Chalfie, 2009*; *Cameron et al., 2010*; *Chen et al., 2010*; *Zelle et al., 2013*). Through manual curation of geneset annotations, we predict that the *Ae. aegypti* genome (*Matthews et al., 2018*) contains 31 *ppk* genes. These are unevenly distributed across its three chromosomes. Whereas chromosome 1 has only three *ppk* genes, chromosome 3 has 22, including three clusters of 3–7 *ppk* genes (*Figure 2A*). The predicted protein products of many *ppk* genes have clear 1-to-1 orthologues in *Drosophila melanogaster* and the malaria mosquito

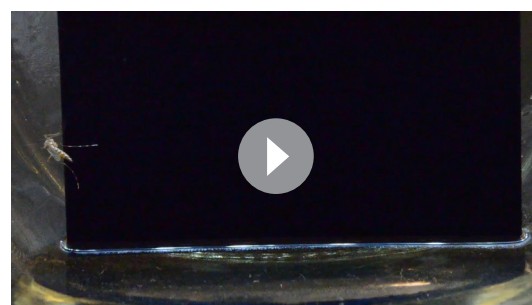

**Video 1.** *Ae. aegypti* mosquitoes use contact with freshwater to guide egg-laying (related to *Figure 1*). This video shows a single female mosquito, 4 days after a blood-meal, in a small chamber with deionized water and a vertically oriented piece of black paper partially submerged in the water. The mosquito directly contacts the liquid with sensory appendages including all legs and proboscis, followed by egg-laying on the moist black paper substrate above the water line. Freshly laid *Ae. aegypti* eggs are initially un-melanized and therefore appear white in this video and in *Figure 1A*.

DOI: https://doi.org/10.7554/eLife.43963.004

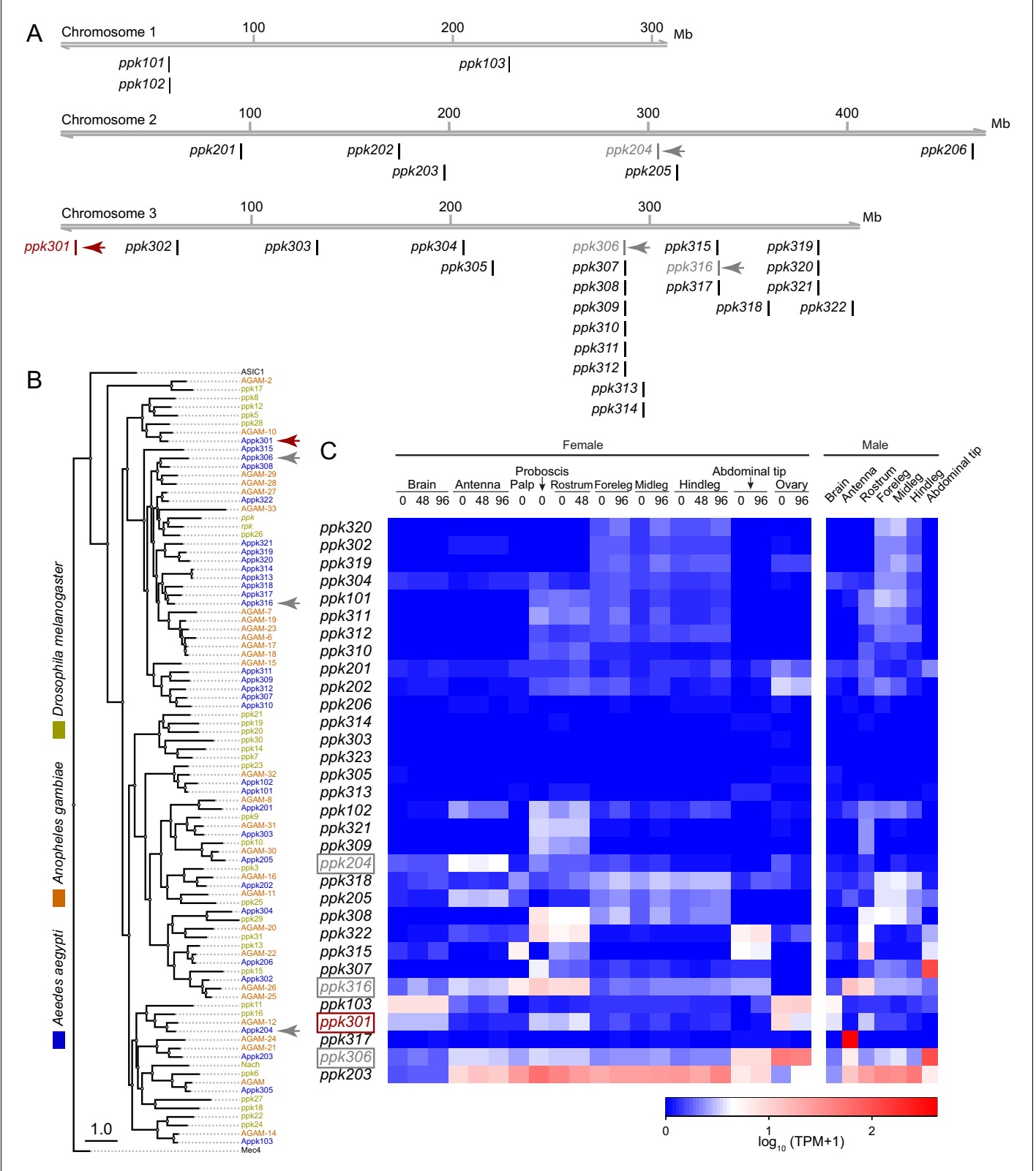

**Figure 2.** Genomic organization and tissue-specific expression of *Ae. aegypti ppk* ion channels. (**A**) Genomic organization of ppk ion channels on the three chromosomes of *Ae. aegypti*. (**B**) Phylogenetic tree of ppk ion channel proteins from *Ae. aegypti, Anopheles gambiae*, and *Drosophila melanogaster*. Scale bar = 1.0 substitution/site. (**C**) Gene expression of *ppk* ion channels across tissues. Data originally generated in a survey of gene

*Figure 2 continued on next page*

*Figure 2 continued*

expression across neural tissues (*Matthews et al., 2016*) were aligned to the AaegL5 genome (*Matthews et al., 2018*). Values represent mean of multiple replicates for each tissue. Genes studied in this paper are marked with arrowheads in A-C.

DOI: https://doi.org/10.7554/eLife.43963.005

*Anopheles gambiae*, while others are in species-specific expansions (*Figure 2B*). Examination of tissue-specific transcript abundance using previously published RNA-seq data (*Matthews et al., 2016*; *Matthews et al., 2018*) reveals a broad range of expression patterns in adult tissues, including in legs, proboscis, and other sensory tissues (*Figure 2C*). In a previous study (*Kistler et al., 2015*), we generated CRISPR-Cas9 mutations in four *ppk* genes (marked with arrowheads in *Figure 2A,B*) that we considered as candidates for controlling egg-laying.

We first measured the ability of these mutant strains to blood-feed and found that all four were attracted to and engorged fully on the blood of a live human host (*Figure 3A*). To test egg-laying behavior, we introduced single blood-fed females of each strain into egg-laying vials containing a small amount of water and a filter paper as an egg-laying substrate. One mutant, *ppk301*, an orthologue of the *Drosophila melanogaster ppk28* low-osmolality sensor (*Cameron et al., 2010*; *Chen et al., 2010*), showed a defect in egg-laying. Fewer than 40% of *ppk301* mutants laid more than 10 eggs (*Figure 3B*). To exclude the possibility that this defect was due to an inability to convert blood into developed embryos, we counted the number of mature eggs in ovaries and confirmed that there was no difference between wild-type and *ppk301* mutants (*Figure 3C*).

To investigate freshwater egg-laying preference in the *ppk* mutants, we introduced single blood-fed female mosquitoes into individual chambers containing two Petri dishes, one filled with freshwater (0 mOsm/kg NaCl) and the other with 200 mOsm/kg NaCl. Even when *ppk301* mutant animals laid eggs, they laid fewer eggs than wild-type on water and more eggs than wild-type on 200 mOsm/kg NaCl (*Figure 3D*). We assayed these four *ppk* mutant strains across a range of NaCl concentrations and found a significant reduction in aversion to salt solution only in the *ppk301* mutant, as measured by the proportion of animals laying eggs primarily on salt solution (*Figure 3D–F*). Together, these data support the conclusion that mutations in *ppk301* disrupt freshwater egg-laying in two distinct ways: by dramatically reducing the drive to lay eggs in suitable, low-salt, substrates and reducing the aversion to concentrations of NaCl that are lethal to their offspring.

If *ppk301* mutant animals fail to sense water, which normally triggers mosquitoes to lay an entire clutch of eggs in a short timespan, we hypothesized that these mutants would show a delay in onset and a reduced egg-laying rate when housed in close proximity to water for many days. To ask if *ppk301* mutant animals will lay eggs on water given sufficient time, we introduced individual blood-fed females into egg-laying vials containing water and monitored the number of eggs laid per female per day over 7 days. While the vast majority of wild-type and *ppk301* heterozygous animals laid all their eggs within the first 2 days of being introduced into egg-laying vials, *ppk301* mutant animals did not. Instead, the mutants showed increased variability in the time of egg-laying initiation and a tendency to spread egg-laying out over many days (*Figure 3G–J*). This slow, sustained, and variable egg-laying behavior on freshwater is consistent with a defect in sensing the water that triggers egg-laying.

If *ppk301* directly senses the osmolality or salinity of liquid, we would expect it to be expressed in the sensory appendages that contact water. At the inception of this project, genetic tools for labeling, monitoring, and manipulating neurons in *Ae. aegypti* did not exist. To address this gap, we developed new genetic tools in the mosquito to label all *ppk301*-expressing neurons and to image neuronal activity at sensory neuron terminals using the genetically encoded calcium sensor GCaMP6s (*Chen et al., 2013*). We adapted an approach in which a T2A 'ribosomal skipping' peptide is used to express multiple independent protein products from a single RNA transcript (*Diao and White, 2012*; *Daniels et al., 2014*). We first tested the efficiency of T2A in *Ae. aegypti* by generating a transgene containing a membrane-targeted mCD8:GFP fusion protein and a nuclear-targeted dsRed:NLS fusion protein separated by T2A and driven from the *Ae. aegypti* polyubiquitin promoter (*Anderson et al., 2010*) (*Figure 4A*). Confocal microscopy revealed complete subcellular separation of the two fluorophores in individual larval body-wall cells (*Figure 4A*). This demonstrates that T2A

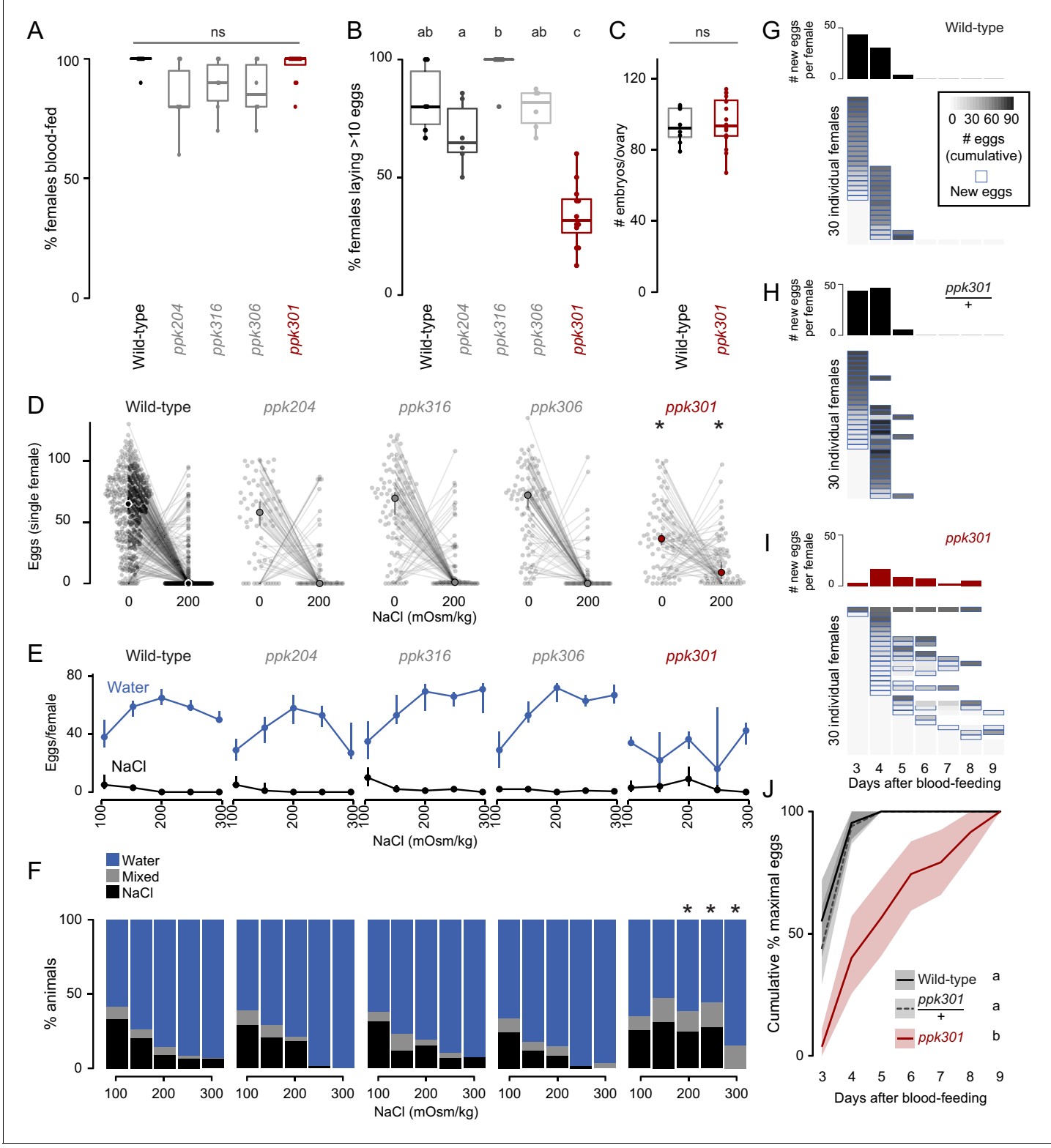

**Figure 3.** Mutations in the *ppk301* ion channel disrupt egg-laying preference and timing. (A, B) Females of indicated genotype who blood-fed (A) or laid >10 eggs given 18 hr to lay (B). n = 6–12 groups/genotype. (C) Embryos/ovary 72 hr post-blood-meal. n = 8–16/genotype. (D–F) Eggs laid by single females of the indicated genotype in the indicated NaCl concentration. Only females laying >10 eggs are presented. In D, lines connect data from individual animals (dots). Median (points) ± 95% confidence interval (bars). n = 65–314 animals/genotype (D) or n = 3–314 animals/genotype (E) per concentration. In F, proportion of animals are binned into three behavioral groups by eggs laid on each substrate. (G–I) Eggs laid in 0 mOsm/kg NaCl

*Figure 3 continued on next page*

*Figure 3 continued*

by single females of the indicated genotype. n = 30 females/genotype. Blue outline indicates days on which new eggs were laid and histograms indicate mean new eggs (per female). (J) Summary of data in G-I. Mean (line) ±95% confidence interval (shaded bars). In B, data labeled with different letters are significantly different from each other p<0.05, in D,F * indicate difference from wild-type p<0.05 (D) p<0.01 (F); ns = not significant, ANOVA followed by Tukey's HSD (A, B, E, H), unpaired t-test (C), and chi-squared test corrected for FDR (F). Boxplots in A-C indicate median and 1st and 3rd quartiles, whiskers extend to 1.5x interquartile range.

DOI: https://doi.org/10.7554/eLife.43963.006

functions efficiently to prevent peptide bond formation in *Ae. aegypti* and can be used as a tool to independently express multiple gene products from a single locus.

To build a flexible system for expressing a diverse array of effector transgenes, we employed the Q-binary expression system for transgene amplification (*Potter et al., 2010*), which has been successfully implemented in *An. gambiae* malaria mosquitoes (*Riabinina et al., 2016*; *Afify et al., 2019*). We used CRISPR-Cas9 with the same guide RNA used to generate the *ppk301* mutant to introduce an in-frame T2A sequence into the *ppk301* locus followed by the QF2 transcriptional activator (*Figure 4B*). This is predicted to cause a loss-of-function mutation in *ppk301* and also express QF2 in all *ppk301*-expressing cells.

We also generated two transgenic QUAS effector strains. The first is a QUAS response element driving the expression of both cytosolic dTomato and GCaMP6s (*15x-QUAS-dTomato-T2A-GCaMP6s*), which allows us to simultaneously label neurons and image their activity with the genetically encoded calcium sensor GCaMP6s (*Chen et al., 2013*) (*Figure 4C*). The second drives expression of membrane-bound GFP (*15x-QUAS-mCD8:GFP*), allowing us to reveal the complete morphology of the neurons in which it is expressed (*Figure 4I*). We looked for expression of the *ppk301>dTomato-T2A-GCaMP6s* reporter in the appendages that contact water during egg-laying (*Figure 4D*) and found that sensory neurons innervating trichoid sensilla in the labellar lobes of the proboscis (*Figure 4E*) were labeled. We also found labeling in the legs, primarily in the distal tarsal segments (*Figure 4F*).

The mosquito central nervous system consists of a brain and a ventral nerve cord (*Ito et al., 2014*; *Smarandache-Wellmann, 2016*; *Court et al., 2017*) (*Figure 4G*). To facilitate our understanding of the neuroanatomy of *Ae. aegypti* in general, and the projection pattern of the *ppk301* driver line in particular, we built a three-dimensional female mosquito brain atlas, in which we identified and named the major neuropils in accordance with nomenclature established by the Insect Brain Name Working Group (*Ito et al., 2014*) (*Figure 4H* and http://mosquitobrains.org). Projections of *ppk301*-expressing sensory neurons in the head extend processes specifically to the subesophageal zone (*Figure 4H,J*). Two nerves enter each hemisphere of the subesophageal zone, arising from the proboscis and the pharynx (*Figure 4J*). Additionally, each leg sends projections into the ventral nerve cord, with nerves running into each neuromere (*Figure 4K*). Both brain and ventral nerve cord innervation patterns are consistent with these neurons mediating taste sensation (*Scott, 2018*). We noted that projections of *ppk301*-expressing neurons are also present in the male brain and ventral nerve cord (*Figure 4—figure supplement 1A–C*), consistent with a role for *ppk301* in behaviors other than egg-laying. Labeling in both males and females was absent in genetic controls expressing either QF2 driver or QUAS effector alone (*Figure 4—figure supplement 1D–K*).

We hypothesized that if *ppk301*-expressing neurons were promoting egg-laying, they should be maximally activated by water, and we set out to test this by functional calcium imaging in response to freshwater and behaviorally relevant concentrations of NaCl. We developed an in vivo calcium imaging preparation with GCaMP6s (*Chen et al., 2013*) in ventral nerve cord sensory afferents of a mosquito presented with water or NaCl solutions on a single foreleg (*Figure 5A,B*). This appendage was chosen because it most frequently contacts water during egg-laying (*Figure 1A*, *Video 1*). We imaged the prothoracic segment of the ventral nerve cord with two-photon microscopy, using a custom fluidics device to deliver and retract liquids to the foreleg and compared responses to different stimuli within individual animals. *ppk301*-expressing neuronal projections in the ventral nerve cord were identified by dTomato expression (*Figure 5C*), which was also used to determine the region of interest for calcium imaging analysis.

We observed low GCaMP6s fluorescence at baseline and an increase in fluorescence in every trial where water was presented (*Figure 5C–E*), with no apparent desensitization across trials (data not

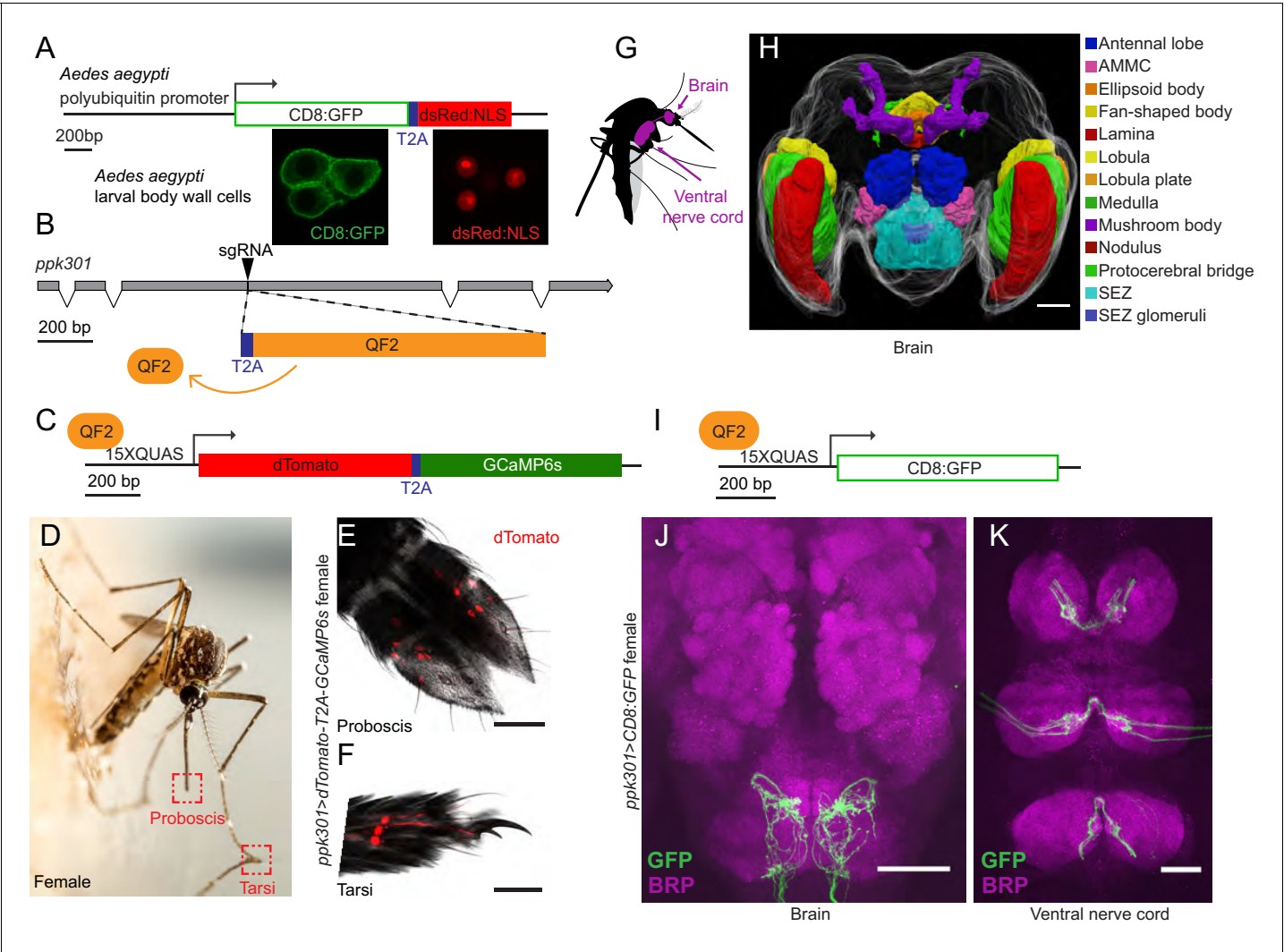

**Figure 4.** In-frame insertion of T2A-QF2 into the *ppk301* locus labels sensory neurons that project to central taste centers. (**A**) Top, construct used to test the efficacy of T2A in *Ae. aegypti*. Bottom, confocal images of three *Ae. aegypti* body wall cells expressing both mCD8:GFP (green) and dsRed: NLS (red). (**B**) Diagram of coding region of *ppk301* locus with exons (gray boxes), introns (connecting lines), and CRISPR-Cas9 gRNA site used to insert T2A (blue) and QF2 (orange). (**C**) Map of *15x-QUAS-dTomato-T2A-GCaMP6s* transgene. (**D**) Female mosquito laying an egg on a moist substrate, highlighting proboscis and tarsal appendages that contact water (red boxes). Photo: Alex Wild. (**E, F**) Confocal image dTomato expression in *ppk301>dTomato-T2A-GCaMP6s* proboscis (**E**) and tarsi (**F**) with transmitted light overlay. (**G**) Cartoon of mosquito adult neural tissues. (**H**) 3D-reconstruction of female *Ae. aegypti* brain. Different regions of the brain are identified by Bruchpilot (Brp) immunostaining and homology to other insects. AMMC = Antennal Mechanosensory and Motor Center, SEZ = subesophageal zone. (**I**) Map of *15x-QUAS-mCD8:GFP* transgene (**J, K**) Expression of *ppk301>mCD8:GFP* in female brain (**J**) and ventral nerve cord (**K**) Scale bars: 50 μm.

DOI: https://doi.org/10.7554/eLife.43963.007

The following figure supplement is available for figure 4:

**Figure supplement 1.** Male expression of ppk301>mCD8:GFP and QF2/QUAS controls for ppk301-QF2 reagent.

DOI: https://doi.org/10.7554/eLife.43963.008

shown). Only the ipsilateral side of the ventral nerve cord innervated by the stimulated leg showed activation (data not shown). In *D. melanogaster*, cells that express the *ppk301* orthologue *ppk28* respond to water but are inhibited by high osmolality solutions including NaCl (*Cameron et al., 2010*; *Chen et al., 2010*; *Jaeger et al., 2018*). The population response in *ppk301*-expressing afferents in the mosquito is functionally different, showing robust responses to water and strong activation by salt solutions (*Figure 5D–F*). To determine whether ppk301 is required for the neuronal responses to water or salt, we performed imaging experiments in a *ppk301* mutant background and

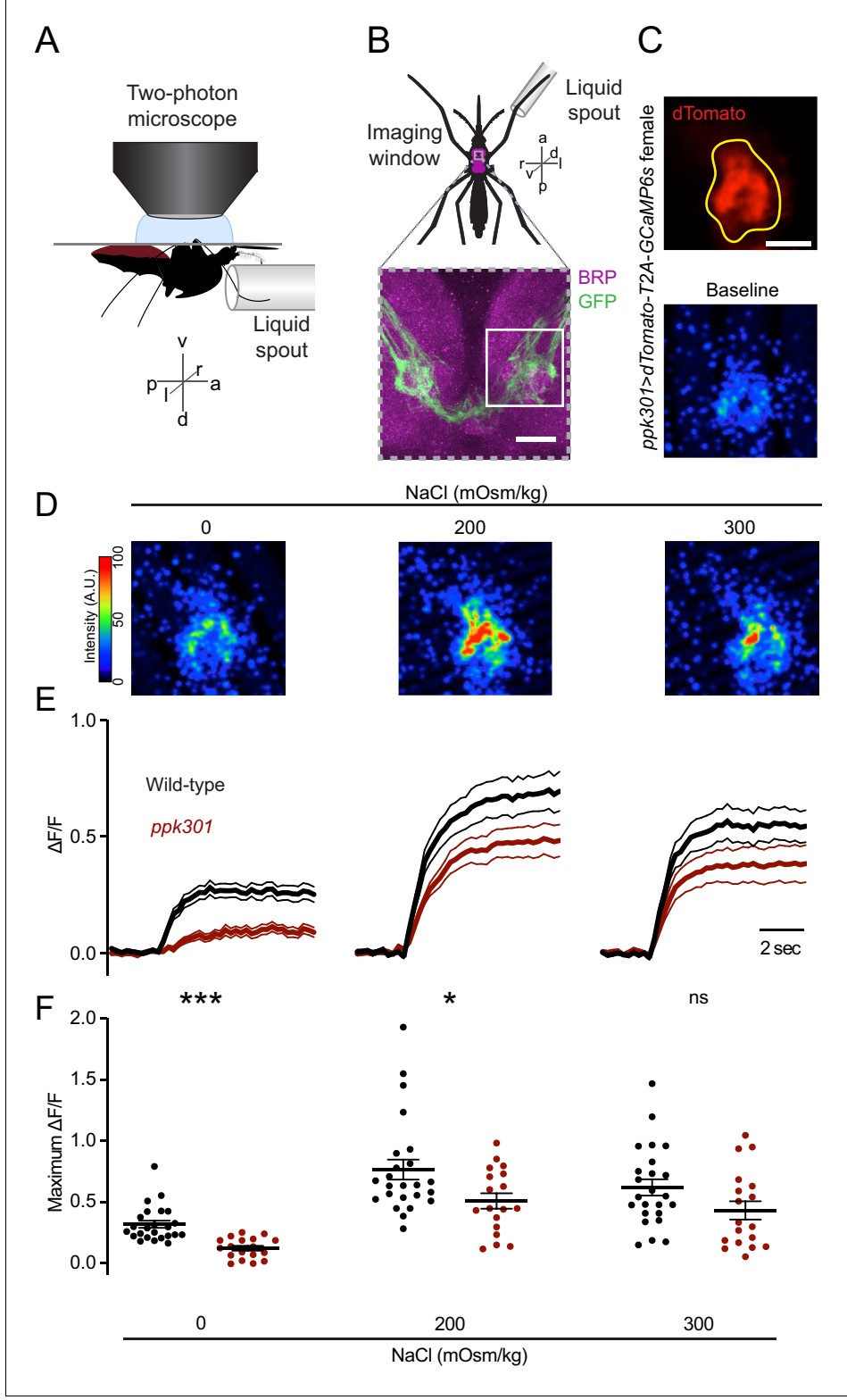

**Figure 5.** *ppk301* neurons respond to water and salt. (**A**) Side view schematic of ventral nerve cord imaging setup. (**B**) Top, ventral view schematic of mosquito during imaging showing ventral nerve cord (magenta) and imaging window (gray box). Bottom, confocal image of the approximate area contained in the imaging window. White box highlights the approximate area in C. Image from *ppk301>mCD8:GFP* animal. Scale bar 10 µm. (**C**) Top, two-photon image of dTomato (red) and region of interest (yellow). Scale bar 20 µm. Bottom: Representative

*Figure 5 continued on next page*

*Figure 5 continued*

GCaMP6s fluorescence at baseline (scale: arbitrary units). (D) Representative GCaMP6s fluorescence increase to indicated NaCl concentrations compared to baseline (scale: arbitrary units). (E) Average GCaMP6s traces from all wild-type and *ppk301* mutant animals presented with three concentrations of NaCl until end of the trial, n = 24 sweeps from eight animals each. Mean (thick lines) ± S.E.M. (thin lines). (F) Summary of data in E comparing the maximum ΔF/F (arbitrary units) for each individual stimulus presentation in wild-type and *ppk301* mutant animals. Lines denote mean and SEM. *p<0.05 ***p<0.001, t-test between genotypes at each concentration with Bonferroni correction.

DOI: https://doi.org/10.7554/eLife.43963.009

found that the response to water is almost entirely abolished in the *ppk301* mutant (*Figure 5D–F*). In contrast, 200 and 300 mOsm/kg NaCl solutions still elicited a strong response in the *ppk301* mutant, albeit with a small reduction in amplitude (*Figure 5D–F*). These data show that *ppk301* is required for the activation of these neurons by freshwater and that a *ppk301*-independent pathway activates these neurons in response to salt. This suggests that mosquito *ppk301*-expressing afferents are either multi-modal (*Zocchi et al., 2017*) and each neuron responds to both water and salt, or that there is a functionally heterogeneous population of *ppk301*-expressing afferents with distinct neurons responding to water and salt.

To test if *ppk301*-expressing neurons are activated by any solution with increased osmolality, we performed imaging experiments with two additional solutes that are not ionic salts, L-serine and D-(+)-cellobiose. We delivered all stimuli at 200 mOsm/kg, the osmolality of NaCl that elicited the peak response. We found that *ppk301*-expressing neurons did not respond more strongly to L-serine or D-(+)-cellobiose than they did to water alone (*Figure 6A–B*). Thus, activity in these neurons does not only track with osmotic pressure. Intrigued by these findings, we performed egg-laying preference assays where mosquitoes were given a choice to lay eggs on freshwater and either L-serine or D-(+)-cellobiose. Although female mosquitoes found 200 mOsm/kg NaCl highly aversive, they did not discriminate between freshwater and L-serine or D-(+)-cellobiose at 200 mOsm/kg (*Figure 6C–D*). Thus, the behavioral aversion to salt is unrelated to osmotic pressure.

The observation that *ppk301*-expressing cells are activated by both water and salt is intriguing because wild-type females avoid laying eggs specifically in high-salt solutions. To understand how female mosquitoes interact with these different stimuli, we monitored real-time behavior of individual females offered either water or 300 mOsm/kg NaCl over 40 min by scoring their contact with liquid and individual egg-laying events. Both wild-type and *ppk301* mutant mosquitoes contacted water, but only wild-type females consistently transformed these touches into egg-laying events (*Figure 7A*). When offered 300 mOsm/kg salt solution, both wild-type and *ppk301* mutant mosquitoes touched liquid, but neither genotype reliably laid eggs (*Figure 7A*). These results are consistent with a model (*Figure 7B*) in which activation of *ppk301* cells by water is a permissive signal for rapid and reliable egg-laying. Animals lacking *ppk301* fail to detect the water activation signal and show delayed and intermittent egg-laying. Activation of *ppk301*-expressing neurons by high salt is not sufficient to drive egg-laying, suggesting that *ppk301*-expressing cells gate egg-laying at low NaCl concentrations, but as NaCl concentrations increase, an independent noxious salt-sensing pathway is recruited that overrides the activation of the tarsal *ppk301*-expressing cells imaged in this study to inhibit egg-laying (*Figure 7B*). We predict mutations that disrupt this noxious salt sensor would yield mosquitoes that show indiscriminate egg-laying on a high-salt substrate.

## Discussion

*Ae. aegypti* mosquitoes preferentially lay eggs in freshwater and avoid saltwater. We annotated the *ppk* ion channel gene family in *Ae. aegypti* and made targeted mutations in several candidate genes, identifying a mutation in *ppk301* that disrupts freshwater egg-laying preference and behavior. *ppk301* is expressed in sensory neurons of the tarsi and proboscis, tissues that contact water during egg-laying. *ppk301*-expressing tarsal sensory neurons are activated by both water and high salt. This response is not driven solely by osmotic pressure, because two other solutes L-serine and D-(+)-cellobiose presented at the same osmolality were indiscriminable from water. In a *ppk301* mutant background, responses to water are almost entirely abolished but responses to salt remain. Together,

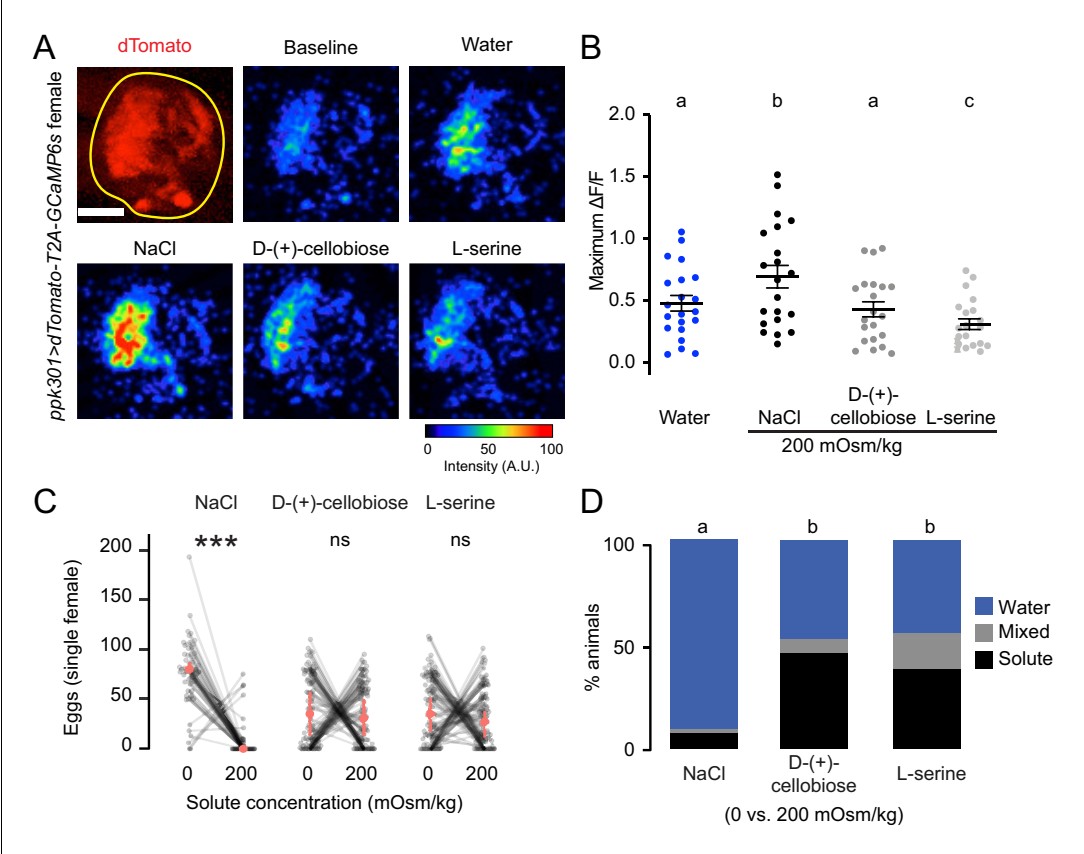

**Figure 6.** *ppk301* neurons are not tuned to osmotic pressure. (**A**) Top left, two-photon image of dTomato (red) and region of interest (yellow). All other panels show representative GCaMP6s fluorescence increase to indicated solutions at 200mOsm/kg compared to baseline (scale: arbitrary units). (**B**) Maximum ΔF/F (arbitrary units) for individual sweeps in wild-type animals. n = 21 sweeps/condition from n = 7 animals. (**C**) Eggs laid by single females of the indicated genotype in the indicated solute concentration. Only females laying >10 eggs are presented, and lines connect data from individual animals. n = 53–79 animals/genotype. (**D**) Proportion of animals are binned into three behavioral groups by eggs laid on each substrate. Scale bar: 10 μm (**B**). In B, data labeled with different letters are significantly different from each other (two-way ANOVA with Tukey's HSD p<0.05). In (**C**) stars denote that the number of eggs laid on water are significantly different than the number of eggs laid on each solute (Paired Wilcoxon rank-sum test corrected for multiple comparisons by the Bonferroni method, ***=p < 0.001). In (**D**), different letters indicate that conditions are different from one another (p<0.05) by a chi-squared test corrected for FDR.
DOI: https://doi.org/10.7554/eLife.43963.010

these data suggest a model in which activation of *ppk301*-expressing neurons by water is required for egg-laying while a *ppk301*-independent pathway encodes aversion to high concentrations of salt (*Figure 7B*). This study identifies an important component of the circuitry regulating egg-laying behavior and provides an entry point into understanding the most important parenting decision a female mosquito makes.

What sensory cues and modalities activate *ppk301*-expressing neurons? Sensory neurons in the *D. melanogaster* proboscis that express the orthologous *ppk28* gene are maximally activated by water but inhibited by dissolved solutes (*Cameron et al., 2010*; *Chen et al., 2010*; *Jaeger et al., 2018*). In contrast, the afferents of the *ppk301*-expressing cells in the mosquito tarsi are not tuned specifically to osmolality but rather encode multimodal responses to liquid (*ppk301*-dependent) and to high concentrations of salt (*ppk301*-independent). Whether or not individual neurons within this population exhibit distinct tuning properties remains to be determined. In addition to the tarsi, *ppk301* is expressed in the proboscis and pharynx. It will be important to evaluate whether these populations of *ppk301*-expressing cells are strictly tuned to water, as are *D. melanogaster ppk28*-expressing labellar sensory neurons, and lack the multimodal responses seen in the tarsal *ppk301*-expressing neurons.

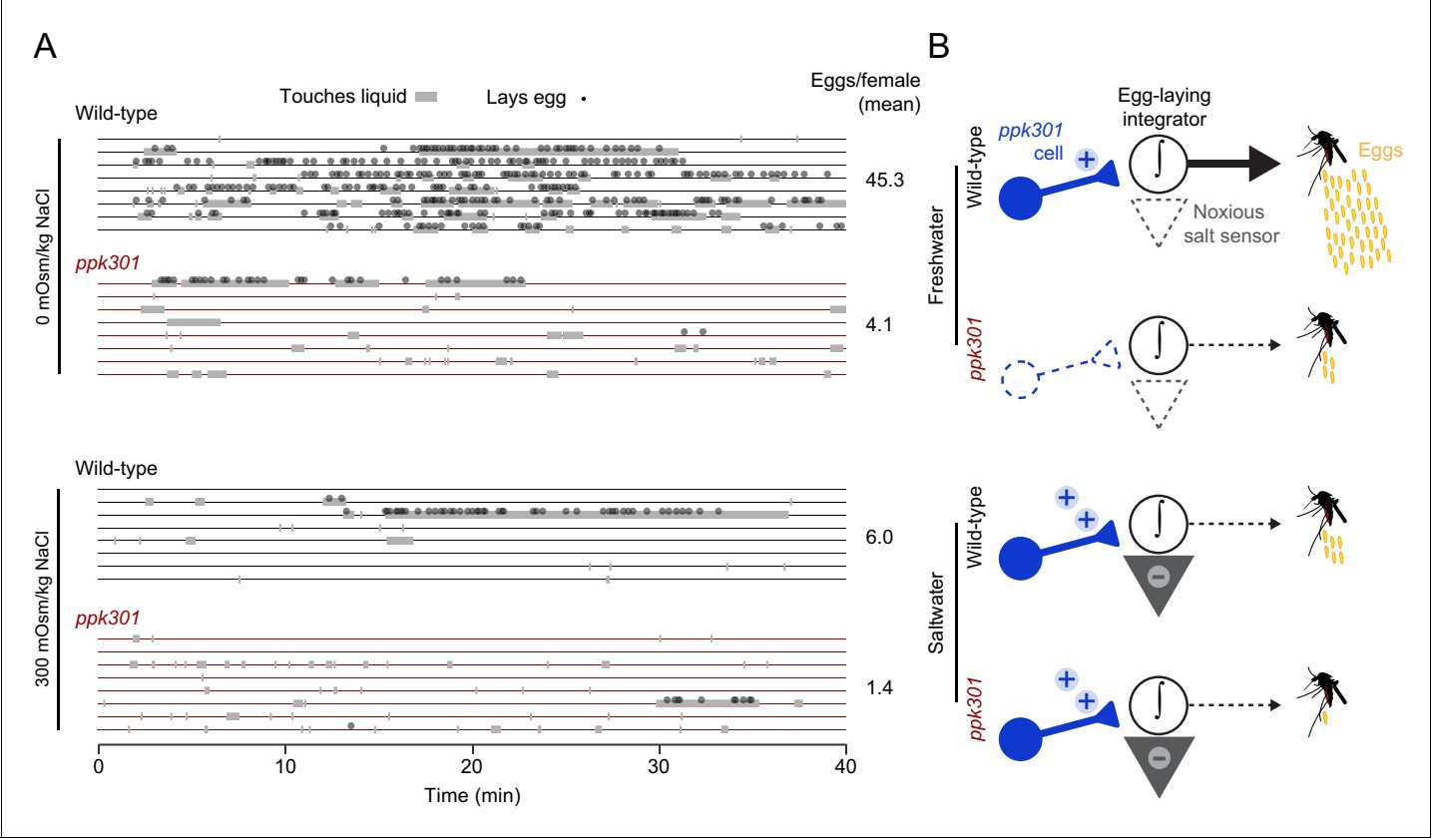

**Figure 7.** *ppk301* gates egg laying, pointing to integration of water and salt cues that drive preference. (**A**) Raster plot of eggs laid (circles) and physical contact with liquid (gray boxes) by eight individual females of the indicated genotype over 40-min observation period in the indicated NaCl solution. (**B**) Model for *ppk301* control of freshwater egg-laying.

DOI: https://doi.org/10.7554/eLife.43963.011

How does *ppk301* regulate egg-laying? We favor a model in which *ppk301*-dependent activation of sensory neurons by liquid provides a permissive signal required for egg-laying, while high concentrations of salt activate noxious salt pathway(s) that can override this signal and prevent egg-laying. Together, these opposing signals are integrated to sculpt egg-laying preference and allow a mosquito to exploit freshwater substrates while preventing eggs from being laid on high-salt substrates that are lethal to their offspring.

It will be interesting to discover the biophysical mechanisms by which *ppk301* neurons respond to water. *ppk301* encodes a single subunit of a trimeric DEG/ENaC channel (*Jasti et al., 2007*). One possibility is that *ppk301* ion channels are homomers that respond to osmotic pressure, similar to *D. melanogaster ppk28* (*Cameron et al., 2010*; *Chen et al., 2010*). Alternatively, *ppk301* could encode a promiscuous subunit that could be part of multiple heteromeric PPK channels with different ligands that account for some of the multimodal tuning properties seen in *ppk301*-expressing tarsal sensory neurons. In this scenario, genetic ablation of *ppk301* could result in a novel set of PPK channels that are composed of the remaining subunits, thereby shifting the composition of channels on the membrane and influencing overall neuronal tuning properties.

Water- and salt-sensing are critically important because all animals must regulate fluid and ion homeostasis for the proper functioning of their physiological systems. Terrestrial insects such as mosquitoes are small and have a large surface-area-to-volume ratio, meaning that they are susceptible to desiccation. Therefore, both male and female adult mosquitoes must seek out liquid for survival. It is not known if sensory systems used for determining the physical properties of liquids during fluid ingestion in male and female mosquitoes are the same as those used by the female for evaluating water quality during egg-laying. Our data show that *ppk301* is expressed in both male and female

appendages, and that neurons expressing *ppk301* project to similar areas of the brain and ventral nerve cord. Studying the effects of *ppk301* mutations on liquid ingestion in adult *Ae. aegypti* will provide insight into whether the same molecules and sensory neurons are being used in distinct behavioral contexts. It will also be interesting to investigate if the activity of *ppk301* neurons is modulated by physiological state, including the presence of developed eggs or elevated thirst due to dehydration or desiccation stress. Previous studies have shown the existence of a canonical 'water' cell in the apical labral chemoreceptor sensilla of *Ae. aegypti* (**Werner-Reiss et al., 1999**), but we do not see *ppk301* expression in this tissue, indicating that there may be additional osmosensitive receptors encoded in the mosquito genome.

A complete cellular and molecular understanding of salt sensation in mosquitoes remains to be elucidated and the 'noxious salt sensor' proposed in our model remains to be discovered. We note that a distributed coding of salt taste in which distinct populations of sensory neurons are modulated by salt is not unprecedented. A recent study in *Drosophila* reveals that in the proboscis, salt can activate or inhibit every class of gustatory receptor neuron (**Jaeger et al., 2018**). Salt is also a key component of blood and understanding the mechanisms of salt sensation during blood-feeding, egg-laying, and other critical mosquito behaviors is an important future goal (**Werner-Reiss et al., 1999**; **Sanford et al., 2013**).

The evolution of egg-laying preference both within *Ae. aegypti* and in other mosquito species is a critical determinant of the types of environments and ecological niches that they are able to exploit. Indeed, some strains of *Ae. aegypti* have begun to exploit brackish water for egg-laying (**Ramasamy et al., 2014**; **de Brito Arduino et al., 2015**). Such an adaptation requires co-evolution of the shifting preference of females for salty water and larval tolerance of high salinity environments into which they are deposited. Investigating the underlying genetic changes that support the coordination of these shifting phenotypes in both larvae and adults will be important for understanding how *Ae. aegypti* adapts to novel breeding sites across the globe, which has greatly increased their capacity as a disease vector.

The advances described here depended critically on new genetic tools we developed that did not previously exist in this non-model insect. These techniques now make it possible to visualize neuronal anatomy and activity within molecularly identified cell types and to identify the genetic and neural circuit substrates of many mosquito behaviors. These behaviors contribute to the spread of deadly pathogens and understanding their underlying biology could contribute to mosquito control efforts. The development of similar tools and reagents in other insect species of public health, agricultural, or ethological interest will broaden our view of insect biology and facilitate comparative studies of the genes and circuits underlying evolutionary adaptations in insects.

## Materials and methods

**Key resources table**

| Reagent type (species) or resource | Designation | Source or reference | Identifiers | Additional information |
|---|---|---|---|---|
| Antibody | anti-Brp (mouse monoclonal) | Developmental Studies Hybridoma Bank | DSHB:NC82 RRID:AB_2314866 | (1:50) |
| Antibody | anti-GFP (rabbit polyclonal) | Life Technologies | Molecular Probes: A11122 RRID:AB_221569 | (1:10,000) |
| Chemical compound, drug | L-serine | Sigma | Sigma:s4500 | |
| Chemical compound, drug | D-(+)-cellobiose | Sigma | Sigma:22150 | |
| Chemical compound, drug | NaCl | Sigma | Sigma:s7653 | |
| Gene (*Aedes aegypti*) | *ppk301* | NA | NCBI:LOC5564307 | previously known as AAEL000582, ppk582 |

*Continued on next page*

*Continued*

| Reagent type (species) or resource | Designation | Source or reference | Identifiers | Additional information |
|---|---|---|---|---|
| Gene (*Ae. aegypti*) | *ppk204* | NA | NCBI:LOC5573846 | previously known as *AAEL010779, ppk10779* |
| Gene (*Ae. aegypti*) | *ppk316* | NA | NCBI:LOC5567201 | previously known as *AAEL000926, ppk00926* |
| Gene (*Ae. aegypti*) | *ppk306* | NA | NCBI:LOC5563925 | previously known as *AAEL014228, ppk14228* |
| Genetic reagent (*Ae. aegypti*) | *ppk301 -/-* | PMID:25818303 | NA | previously generated and referred to as *AAEL00582 ECFP* in *Kistler et al., 2015* (PMID:25818303). Contains an insertion of a poly-ubiquitin promoter-driven *ECFP* into the coding sequence of *ppk301* |
| Genetic reagent (*Ae. aegypti*) | *ppk204 -/-* | PMID:25818303 | NA | previously generated and referred to as *AAEL010779 -/-* in *Kistler et al., 2015* (PMID:25818303). Contains a 150 bp deletion in the coding sequence of *ppk301* with a corresponding integration of a cassette with triple-stop codons. |
| Genetic reagent (*Ae. aegypti*) | *ppk316 -/-* | PMID:25818303 | NA | previously generated and referred to as *AAEL00926-/-* in *Kistler et al., 2015* (PMID:25818303). Contains an 86 bp deletion in the coding sequence of ppk316 with a corresponding integration of a cassette with triple-stop codons and several downstream SNPs. |
| Genetic reagent (*Ae. aegypti*) | *ppk306 -/-* | PMID:25818303 | NA | previously generated and referred to as *AAEL014228-/-* in *Kistler et al., 2015* (PMID:25818303). Contains a 106 bp deletion in the coding sequence of *ppk306* with a corresponding integration of a cassette with triple-stop codons. |
| Genetic reagent (*Ae. aegypti*) | *ppk301-T2A-QF2* | this paper | NA | T2A-QF2 knockin into the *ppk301* locus, described in Materials and methods. |

*Continued on next page*

*Continued*

| Reagent type (species) or resource | Designation | Source or reference | Identifiers | Additional information |
|---|---|---|---|---|
| Genetic reagent (*Ae. aegypti*) | *Pub-mCD8:GFP-T2A-dsRed:NLS-SV40* | this paper | NA | *Ae. aegypti* poly-ubiquitin promoter driving expression of membrane tethered GFP and nuclear dsRed separated by a T2A peptide. Described in Materials and methods |
| Genetic reagent (*Ae. aegypti*) | *15x-QUAS-mCD8-GFP* | this paper | NA | QUAS reporter strain driving expression of membrane-targeted GFP. Produced by transforming wild-type *Ae. aegypti* with plasmid RRID:Addgene_104878 |
| Genetic reagent (*Ae. aegypti*) | *15x-QUAS-dTomato-T2A-GCaMP6s* | this paper | NA | QUAS reporter strain driving expression of cytosolic dTomato and GCaMP6s separated by a T2A peptide. Described in Materials and methods |

## Mosquito rearing and maintenance

*Aedes aegypti* wild-type laboratory strains (Liverpool) were maintained and reared at 25–28°C, 70–80% relative humidity with a photoperiod of 14 hr light: 10 hr dark as previously described (*DeGennaro et al., 2013*). Adult mosquitoes were provided constant access to 10% sucrose. For routine strain maintenance, animals were blood-fed on live mice or human subjects. Adult females of all genotypes were blood-fed on a single human subject during mutant generation, for behavioral assays, and for calcium imaging. Blood-feeding procedures with live hosts were approved and monitored by The Rockefeller University Institutional Animal Care and Use Committee (IACUC protocol 15772) and Institutional Review Board, (IRB protocol LV-0652). Human subjects gave their written informed consent to participate.

## Blood-feeding for egg-laying behavior

All *Ae. aegypti* mosquitoes used in egg-laying assays were housed in mixed-sex cages and were between 7 and 14 days old. Female mosquitoes were blood-fed by giving them direct access to a human limb introduced in or on the mesh wall of a cage. Fully engorged female mosquitoes were selected within 24 hr of blood-feeding and housed in insectary conditions with ad libitum access to 10% sucrose before being introduced into behavior assays. Unless otherwise indicated, assays were performed beginning 96 hr post-blood-meal and continuing for ~18 hr.

## Mesh barrier assay

10 blood-fed female mosquitoes were introduced by mouth aspiration into a standard BugDorm rearing cage (30 cm$^3$) containing a 10% sucrose wick. Two egg-laying cups were present in each cage, each containing deionized water and a 55 mm-diameter partially submerged Whatman filter paper (Grade 1; 1001–055). In one of the cups, a metal mesh barrier placed ~1 cm above the water line prevented direct access of the mosquitoes to liquid, although the filter paper remained partially submerged and moist. Mosquitoes were allowed to lay eggs overnight (18 hr), after which the filter papers were dried and eggs counted.

## Seawater preference assay

Artificial seawater was mixed according to an established recipe (*Kester et al., 1967*) with lab-grade chemicals from Sigma-Aldrich. Specific dilutions were made, by volume, with deionized water. Single

blood-fed females were introduced into a chamber comprised of a length of transparent acrylic tubing (inner diameter 9.525 cm) cut to 12 cm in height with a wire mesh grid glued to one end as a ceiling containing a two-sector 90 mm divided Petri dish with 10 mL of deionized water on one side and 10 mL of a specific dilution of artificial seawater on the other. As an egg-laying substrate, a 1 cm tall strip of seed germination paper (Anchor Paper; SD7615L) was wrapped around the outer diameter of each half of the Petri dish, partially submerged in the liquid. Animals were introduced by a mouth aspirator through a hole in the top of the chamber, which was then plugged with a cotton ball. Containers were stored under insectary conditions and mosquitoes were allowed to lay eggs overnight (~18 hr), after which the paper strips were dried and eggs counted.

### Seawater survival assay

Wild-type mosquitoes were hatched in 'hatch broth' consisting of deoxygenated water containing ground Tetramin fish food. Approximately 1 day after hatching, 20 larvae were transferred into a small plastic cup containing 25 mL of a specific dilution of artificial seawater. Cups were examined each day, dead larvae removed, and ground Tetramin pellets added for food as needed. Animals successfully completing the transition to pupal stage by 8 days post-hatch were scored as surviving offspring.

### Single-female modular egg-laying assay

A multi-animal egg-laying assay was created out of sheet acrylic, comprising modular trays each with 14 single-animal chambers. Each chamber comprised two angled 50 mm Petri dishes each containing 2 mL of liquid and a 47 mm diameter filter paper (Whatman, Grade 1 Qualitative Filter Paper).

We next developed a standardized imaging setup to automatically count eggs from each Petri dish. Each dish was placed onto a LED light panel (SuperbrightLEDs.com item #2020) and images were captured with a Raspberry Pi and associated camera. Images were thresholded and pixels counted for each dish of each chamber. The number of eggs corresponding to each image was determined by dividing the average pixel value of eggs determined from a manually-counted set of 20 test images. The concordance between manual counting and automatic pixel-based counting was $r^2 = 0.96$. A parts list and schematics of the egg-laying chambers and the image acquisition and thresholding code can be found on http://github.com/VosshallLab/MatthewsYoungerVosshall2018/ (*Vosshall Lab, 2019*; copy archived at https://github.com/elifesciences-publications/ MatthewsYoungerVosshall2018).

For studies of the effect of NaCl on egg production (*Figure 1F*), each dish was filled with the same concentration of NaCl. The osmolality of each solution was measured using a Wescor model 5520 vapor pressure osmometer. Blood-fed female mosquitoes were cold anesthetized and single animals introduced into each chamber by mouth aspiration. Animals were allowed to lay for 18 hr and dishes were imaged to calculate egg numbers. For two-choice assays (*Figure 3D–F*), assays were performed identically, except that the two dishes contained deionized water and a NaCl solution of a specific osmolality. In *Figure 6C–D*, the two dishes contained deionized water and either 200mOsm/kg L-serine (Sigma S4500) or D-(+)-cellobiose (Sigma 22150). The position of each solution was varied for each chamber. Experiments were performed anonymized to genotype and data included only for those animals who laid more than 10 eggs.

### NaCl survival assay

Wild-type mosquitoes were hatched in hatch broth. Approximately 1 day after hatching, 20 larvae were transferred into a small plastic cup (VWR HDPE Multipurpose Containers; H9009-662) containing 25 mL of a specific concentration of NaCl, prepared as above. Cups were examined each day, dead larvae removed, and ground Tetramin pellets added for food as needed. The number of pupae and larvae remaining alive were counted each day, and cumulative survival was calculated for these offspring.

### Egg-laying timing: vial assay

To measure the timing of egg-laying across days, individual blood-fed mosquitoes were introduced into egg-laying vials 48 hr after a blood-meal. Mosquitoes were transferred to a fresh vial every 24

hr and eggs from the previous day were counted and recorded. Experiments were performed anonymized to genotype.

## Egg-laying timing: culture flask assay

To measure the timing of egg-laying and liquid touching, single animals were introduced into a 50 mL cell culture flask containing 10 mL of either deionized water or 300 mOsm/kg NaCl, and a 1' x 2' strip of seed germination paper, partially submerged. Animals were video recorded for 40 min using a Nikon D7000 SLR with a macro lens. Four flasks were recorded simultaneously, with each set containing 1 replicate of the following conditions: wild-type, 0 mOsm/kg NaCl; wild-type, 300 mOsm/kg; *ppk301*, 0 mOsm/kg; *ppk301*, 300 mOsm/kg. Videos were manually scored for physical contact with liquid and the appearance of freshly laid eggs. Data on physical touches were recorded in 5 s intervals, with each interval scored as 'touch' if a single frame revealed physical contact between any appendage of the mosquito and the liquid. Videos were scored anonymized to genotype and condition.

## Curation and expression estimates of *ppk* gene family in *Ae. aegypti*

To identify members of the *ppk* gene family in *Ae. aegypti* and *An. gambiae*, we performed two complementary analyses: 1) using *D. melanogaster* ppk sequences as queries, we performed BLASTp against all translated protein-coding genes identified in AaegL5 (GCF_002204515.2_AaegL5.0_protein.faa) and 2) ran interproscan v5.29.68.0 (*Finn et al., 2017*) against the same database of translated protein-coding genes. We took all genes that were reciprocal best hits with the *D. melanogaster* ppk family via blastp and were annotated by interproscan as 'Amiloride-sensitive sodium channel.' We repeated this analysis for the *An. gambiae*, PEST strain geneset downloaded from Vectorbase (Anopheles-gambiae-PEST_PEPTIDES_AgamP4.9.fa). We renamed *Ae. aegypti ppk* genes by giving them a three-digit identifier corresponding to their chromosomal position. The first digit represents the chromosome (1, 2, or 3), while the next two digits represent its relative position on that chromosome from the left (p) arm to the right (q) arm according to coordinates found on NCBI.

We next built a phylogenetic tree to visualize the relationship between these genes across these three species. To do this, we selected the longest single isoform for genes predicted to encode multiple protein-coding isoforms and performed multiple sequence alignment using clustal-omega v1.2.3 (*Sievers et al., 2011*), including a human ASIC channel and *Caenorhabditis elegans* MEC4 sequence for comparison. A maximum-likelihood-estimate phylogenetic tree was constructed using PhyML v3.0 (*Guindon et al., 2010*) with default parameters and 100 bootstrap iterations, and manually re-rooted on MEC4 for presentation. A table of all genes and sequences incorporated in this analysis, with previous accessions, is presented in the *Supplementary file 1*.

## Transcript abundance estimates of *Ae. aegypti ppk* genes

To visualize transcript abundance of each predicted *ppk* gene across tissues, we utilized published data (*Matthews et al., 2016*; *Matthews et al., 2018*). Heatmaps were generated as described (*Matthews et al., 2018*) and presented as $\log_{10}$(TPM +1) of the mean expression value for all replicates of the indicated tissue using the heatmap.2 function of the gplots v3.0.1 (*Warnes et al., 2016*) package in R v3.5.0 (2017).

## Generation of mutant and transgenic mosquito strains

*ppk* loss-of-function alleles were generated and described previously (*Kistler et al., 2015*). All new strains generated in this paper were injected into wild-type Liverpool embryos at the Insect Transformation Facility at the University of Maryland.

PUb-mCD8:GFP-T2A-dsRed:NLS-SV40 was generated by Gibson assembly, using the following fragments: Plasmid backbone and MOS arms from a standard transformation vector (*pMos-3xP3-dsRed*); *dsRed* open reading frame amplified from the same vector by polymerase chain reaction (PCR); *mCD8-GFP* open reading frame PCR-amplified from a synthesized vector (Genscript); *Ae. aegypti PUb* promoter PCR-amplified from *PSL1180-HR-PUb-ECFP* (Addgene #47917). T2A and NLS sequences were added through PCR. 1000 embryos were injected with the plasmids and a Mos helper plasmid. Two stable lines were recovered with qualitatively similar expression patterns.

*ppk301-T2A-QF2* was generated through CRISPR-mediated homologous recombination into the endogenous locus. We initially attempted to use constructs with gene-specific promoter fragments to drive transgenic constructs, including the broadly expressed polyubitiquin promoter (*Anderson et al., 2010*), but these either did not express at all or were expressed sporadically and mostly in non-neuronal cells (*Bui et al., 2018*). We also had no success with the Gal4/UAS (*Brand and Perrimon, 1993*) system in *Ae. aegypti*. We therefore developed techniques to use homologous recombination to knock QF2 into the *ppk301* locus.

*ppk301-T2A-QF2* was generated with a sgRNA targeting exon 2 of the *ppk301* locus (*ppk301-sgRNA-1*, target sequence with PAM underlined: GGTTGGCAGTTGAGTCC**CGG**). sgRNA DNA template was prepared by annealing oligonucleotides as described (*Kistler et al., 2015*). In vitro transcription was performed using HiScribe Quick T7 kit (NEB, E2050S) following the manufacturer's directions and incubating for 2 hr at 37˚C. Following transcription and DNAse treatment for 15 min at 37˚C, sgRNA was purified using RNAse-free SPRI beads (Ampure RNAclean, Beckman-Coulter A63987), and eluted in Ultrapure water (Invitrogen, 10977–015).

The donor plasmid was constructed by Gibson assembly using the following fragments: homology arms of ~1 kb on either side of the Cas9 cut site (two base pairs were deleted from the left arm immediately preceding the T2A to maintain the open reading frame); a fragment containing *T2A-QF2-SV40* and *3xP3-dsRed*, PCR-amplified from a vector derived from *pBac-DsRed-ORCO_9kb-Prom-QF2* (gift of Chris Potter, Addgene #104877); a *pUC57* vector backbone digested with EcoRI and HindIII. Clones were sequenced verified and midiprepped using an endotoxin free midiprep kit (Machery-Nagel) and eluted in Ultrapure nuclease-free water (Invitrogen). 2000 embryos were injected with a mixture of 300 ng/µL Cas9 protein (PNA Bio), 650 ng/µL dsDNA plasmid donor, and 40 ng/µL sgRNA. The progeny of 96 surviving G0 females were screened. Six potential founders were isolated, and one was verified to have a complete and in-frame insertion by PCR with the following primers (Forward 5' GTGAGGGTGGTGTCGAATTAACTCTT3', Reverse 5'GTTAGGTCA-GAGGTATCCCTGAACAT3').

*15x-QUAS-mCD8-GFP* was generated from an existing plasmid (a kind gift from Chris Potter, Addgene #104878). Embryos were injected with the plasmid and a PBac helper plasmid. Two independent lines were recovered.

*15x-QUAS-dTomato-T2A-GCaMP6s* was generated by Gibson assembly of the following PCR-amplified fragments: Plasmid backbone and Mos arms from *PUb-mCD8:GFP-T2A-dsRed:NLS-SV40* (described above); *15x-QUAS from pBAC-ECFP-15xQUAS_TATA-SV40* (Addgene #104875, gift of Chris Potter); *dTomato-T2A-GCaMP6s* PCR-amplified from a vector synthesized (Genscript). Embryos were injected with the construct and a PBac helper plasmid. Two independent lines were recovered.

## Brain immunostaining

Dissection of adult brains and immunostaining was modified from previously used protocols (*Siju et al., 2008*; *Riabinina et al., 2016*). 6–14 day-old mosquitoes were anesthetized on ice. Heads were carefully removed from the body by pinching at the neck with sharp forceps. Heads were placed in a 1.5 mL tube for fixation with 4% paraformaldehyde, 0.1 M Millonig's Phosphate Buffer (pH 7.4), 0.25% Triton X-100, and nutated for 3 hr. Brains were then dissected out of the head capsule in ice cold $Ca^{+2}$-, $Mg^{+2}$-free phosphate buffered saline (PBS, Lonza, 17-517Q) and transferred to a 24-well plate. All subsequent steps were done on a low-speed orbital shaker. Brains were washed in PBS containing 0.25% Triton X-100 (PBT) at room temperature six times for 15 min. Brains were permeabilized with PBS, 4% Triton X-100, 2% Normal Goat Serum for ~48 hr (two nights) at 4˚C. Brains were rinsed once then washed with PBT at room temperature six times for 15 min. Primary antibodies were diluted in PBS, 0.25% Triton X-100, 2% Normal Goat Serum for ~48 hr (two nights) at 4˚C. The primary antibodies used in this experiment were anti-dmBrp (mouse; 1:50: NC82, DSHB) to label the synaptic neuropil and anti-GFP (Rabbit: 1:10,000; A11122, Life Technologies). Brains were rinsed once then washed in PBT at room temperature six times for 15 min. Secondary antibodies were diluted in PBS, 0.25% Triton X-100, 2% Normal Goat Serum for ~48 hr (two nights) at 4˚C. The primary antibodies used in this experiment were anti-mouse-Cy5 (1:250; Life Technologies A-10524) and anti-Rabbit-488 (1:500; Life Technologies A-11034). Brains were rinsed once then washed in PBT at room temperature six times for 15 min. Brains equilibrated overnight in Vectashield (Vector Laboratories H-1000) and were mounted in Vectashield.

## Ventral nerve cord immunostaining

Six- to 14-day-old mosquitoes were anesthetized on ice and the bodies were carefully removed from the heads by pinching at the neck with sharp forceps. The bodies were placed in a 1.5 mL tube for fixation with 4% paraformaldehyde, 0.1M Millonig's Phosphate Buffer (pH 7.4), 0.25% Triton X-100, and nutated for 3 hr. Ventral nerve cords were dissected out of the body in ice cold PBS and transferred to a 24-well plate. All subsequent steps were the identical to the brain immunostaining protocol described above.

## ppk301>mCD8:GFP immunostaining

*ppk301>mCD8:GFP* expression was visualized in brains and ventral nerve cords using a Zeiss Inverted LSM 880 laser scanning confocal with a 25x/0.8 NA immersion-corrected objective. Glycerol was used as the immersion medium to most closely match the refractive index of the mounting medium Vectashield. Brains were imaged at 2048 × 2048 pixel resolution in X and Y with 0.5 µm z-steps for a final voxel size of 0.1661 × 0.1661×0.5 µm. Ventral nerve cords were imaged at 1024 × 1946 pixel resolution in X and Y with 0.5 µm z-steps for a final voxel size of 0.3321 × 0.3321×0.5 µm ventral nerve cord images were tiled and stitched with 10% overlap. Confocal images were processed in ImageJ (NIH).

   3xP3 was used as a promoter to express fluorescent markers for transgene insertion, and care was taken to distinguish 3xP3 expression from the expression of the *ppk301-QF2* driver line. 3xP3 labels the optic lobes, as well as some cells in the dorsal brain. *Figure 4J,K* and *Figure 4—figure supplement 1B,D,F,H,J* were cropped to remove 3xP3 expression. In some cases, we saw projections from the optic lobes that traversed the brain. These were sometimes seen in the driver alone and effector alone controls, both of which contain a 3xP3 marker transgene. We speculate that these projections derive from cells marked by 3XP3 expression and are unrelated to *ppk301* expression. These patterns of 3xP3 are worth noting, but due to the distance from the subesophageal zone and ventral nerve cord, they do not have an impact on the results of this study. In some brains, we observed faint unilateral projections from the subesophageal zone to the antennal lobe. We examined eight female brains and eight male brains for these projections and saw these projections in 5/8 brains. In males we did not see this projection in any of the eight brains examined.

## Reference brain 3D-Reconstruction

A reference *Ae. aegypti* brain from a 7 day old wild-type female mosquito was fixed and immunostained for Brp as described above. It was imaged as described above except with 1 µm voxels and as a tiled scan with 10% overlap. Thirty-nine brains were analyzed and the most complete and symmetric brain was chosen to serve as the template. Blind deconvolution was performed with AutoquantX3 software. The female brain was manually annotated using the segmentation and 3D reconstruction software ITK-SNAP. Regions were identified by the anatomical boundaries defined by Brp staining, and by homology to other insects, and named in accordance with revised insect brain nomenclature standards (*Ito et al., 2014*). Structures without clear boundaries were excluded. A surface mesh of each region was exported into the data analysis and visualization software ParaView, which was used to generate the 3D reconstruction shown in *Figure 4H*.

## Mosquitobrains.org

The reference brain raw data and reconstruction are available on the website http://mosquitobrains.org. We have displayed this data as a brain atlas by creating an online 'Brain Browser' tool where users can scroll through the brain and highlight different regions. *Ae. aegypti* brains that are stained with the synaptic protein Brp can be warped onto this standard reference *Aedes aegypti* brain using the python code ClearMap (*Renier et al., 2016*). To use ClearMap, all images must be acquired with square voxels. The standard brain was imaged with 1 µm voxel size. To use it as a template, all images must either be taken at this resolution, or down sampled to a final voxel size of 1 µm. Additional channels can be warped and registered onto the reference brain, provided that one channel is imaged as described above.

## T2A test construct validation

The previously characterized *Ae. aegypti* polyubiquitin promoter (Pub) (*Anderson et al., 2010*) was used to drive expression of both a membrane-bound variant of GFP (mCD8:GFP) and a nuclear localization sequence fused to dsRED by separating these genes with the T2A ribosomal skipping element (*Diao and White, 2012*; *Daniels et al., 2014*). This construct was expressed in a few cells in the adult brain, as well as in larval body-wall cells. We focused our analysis on the body-wall cells because their flat and compact shape made them amenable to examining the expression of our membrane and nuclear proteins.

Larvae were dissected by pinning the head and the tail to Sylgard plates (DowDupont) with insect pins, and a long longitudinal cut was made along the dorsal surface of the body wall. The body wall was filleted open with four additional insect pins, and the organs were removed. The animals were fixed for 25 min in 4% paraformaldehyde, 0.1 M Millonig's Phosphate Buffer (pH 7.4), 0.25% Triton X-100 at room temperature, and then rinsed 3 times in PBS. Larvae were transferred to a 1.5 mL tube and washed with PBT 6 times for 15 min while nutating at room temperature. Larvae were then incubated in Vectashield with DAPI overnight and mounted in Vectashield for imaging. Larvae were imaged on a Zeiss Inverted LSM 880 laser scanning confocal with a 40X/1.2 NA oil immersion objective. The cells were imaged at a resolution of 2048 × 2048 pixels in a single confocal slice for a pixel size of 0.0692 × 0.0692 μm. Images were processed in ImageJ (NIH).

## Visualization of *ppk301*-expressing cells in appendages

To visualize sensory neuron cell bodies in *ppk301-T2A-QF2, 15xQUAS-dTomato-T2A-GCaMP6* animals we dissected live sensory tissues using fine forceps, dipped in cold methanol for ~5 s, and mounted on a slide in glycerol. Appendages were viewed on a Zeiss Inverted LSM 880 laser scanning confocal with a 25x/0.8 NA immersion corrected objective at a resolution of 1024 × 1024 pixels and a voxel size of 0.2076 μm x 0.2076 μm x 1 μm.

## Two-photon calcium imaging of female ventral nerve cord

Calcium imaging was performed on an Ultima IV two-photon laser-scanning microscope (Bruker Nanosystems) equipped with galvanometers and illuminated by a Chameleon Ultra II Ti:Sapphire laser (Coherent). GaAsP photomultiplier tubes (Hamamatsu) were used to collect emitted fluorescence. Images were acquired with a 60X/1.0N.A. Long Working Distance Water-Immersion Objective (Olympus) at a resolution of 256 × 256 pixels.

Calcium imaging experiments were performed on female mosquitoes that were 7–11 days post-eclosion. Mosquitoes were fed a human-blood-meal 96–108 hr prior to imaging and were not giving access to an egg laying substrate so that they were gravid at the time of imaging. Gravid females were anesthetized at 4°C for dissection. The wings were removed and the mosquito was fixed to a custom Delrin plastic holder with UV-curable glue (Bondic). The mosquito was inserted into a hole in the holder, such that the ventral thorax, including all coxae, were exposed above the surface of the holder, with the rest of the mosquito below. The mosquito was secured with a few points of glue (Bondic) on the abdomen, thorax and head. The leg that was presented with water remained free of glue to prevent damage to the tissue. Once the mosquito was secured to the plate, one of the forelegs was inserted into a small diameter tube that was secured to the bottom of the plate and could later be attached to the fluidics apparatus used for stimulus delivery (described below). The top of the dish was then filled with external saline. The recipe we used is based *D. melanogaster* imaging saline:103 mM NaCl, 3 mM KCl, 5 mM 2-[Tris(hydroxymethyl)methyl]−2-aminoethanesulfonic acid (TES), 1.5 mM $CaCl_2$, 4 mM $MgCl_2$, 26 mM $NaHCO_3$, 1 mM $NaH_2PO_4$, 10 mM trehalose, 10 mM glucose, pH 7.3, osmolality adjusted to 275 mOsm/kg). The coxae were gently spread from the midline and secured in dental wax. The cuticle was removed above the prothoracic ganglia using very sharp forceps. Opaque non-neural tissue, primarily fat cells and muscle, was removed if they obstructed the ventral nerve cord. Great care was taken not to damage the *ppk301*-expressing nerves running from the legs into the ventral nerve cord. These run up the posterior-ventral region of the leg and are extremely superficial. dTomato fluorescence was examined before imaging to verify that the nerves were intact. If an animal did not respond to water, 200 mOsm/kg, or 300 mOsm/kg NaCl it was discarded.

The preparation was secured to the stage using a custom laser cut acrylic holder. A single plane through the center of the prothoracic neuropil was scanned at 4.22 fps with a 920 nm excitation wavelength imaged through a 680 nm shortpass infrared (IR) blocking filter, a 565 nm longpass dichroic and 595/50 nm or 525/70 nm bandpass filters. GCaMP6s and dTomato emission was collected simultaneously for 70 frames per trial. Each concentration was delivered at least three times per animal, and each animal was exposed to a series of either 0, 200, and 300 mOsm/kg NaCl, or 200 mOsm/kg NaCl, 200 mOsm/kg L-serine, 200 mOsm/kg D-(+)-cellobiose and freshwater. All wild-type and PPK301 mutant animals were reared and imaged in parallel.

Imaging remained stable during the duration of the imaging session in all animals that were included in this study. We did not notice a decrease in the response to stimuli over time. Before beginning the experiments using multiple concentrations of salt, we imaged animals with repeated water delivery, and saw no desensitization to the response to water over 10 presentations (data not shown).

## Liquid delivery

Liquids were delivered to a single foreleg of the mosquito using a custom built low-volume fluidics device. Piezoelectric diaphragm micropumps and their controller (Servoflo) were run by an Arduino using a code written for this purpose. A custom manifold was milled for small volume liquid delivery. The mosquito was illuminated with an IR light and liquid delivery was monitored using an IR camera. The liquid coated the tarsal and tibial segments and was retracted immediately after imaging each sweep. We waited at least 1 min between each trial. We only observed responses in the prothoracic region ipsilateral to the stimulus delivery.

## Data analysis

All image processing was done using FIJI/ImageJ (NIH). Further processing was done using Excel and Prism (GraphPad). Regions of interest were selected based on the dTomato fluorescence intensity and used for analysis of GCaMP6s signal. All traces with motion, as determined by dTomato fluorescence instability, were discarded. A Gaussian blur with a sigma value of 1 was performed on the GCaMP6s signal. In the calculation of $\Delta F/F$, six frames were averaged before stimulus presentation to determine the baseline fluorescence. To determine $F_{max}$, the average of 3 frames at the peak after stimulus delivery was determined for each sweep.

## Statistical analysis

All statistical analyses were performed using Prism (GraphPad) or R version 3.5.0 (*R Development CoreTeam, 2017*). Data collected as percentage of total are shown as median with interquartile range and data collected as raw value are shown as mean ± SEM or mean ± SD. Details of statistical methods are reported in the figure legends.

## Data availability

All plotted data (with the exception of raw video files) are available in *Supplementary file 1*, and behavior assay schematics and egg-laying counting image analysis scripts can be found at https:// github.com/VosshallLab/MatthewsYoungerVosshall2018. (*Vosshall Lab, 2019*; copy archived at https://github.com/elifesciences-publications/MatthewsYoungerVosshall2018). Plasmids will be made available from Addgene.

## Acknowledgements

We thank Raphael Cohn, Emily J Dennis, Andreas Keller, Kevin J Lee, Gaby Maimon, Lindy McBride, Vanessa Ruta, Vikram Vijayan, Nilay Yapici, and members of the Vosshall Lab for comments on the manuscript. We thank the following research assistants, and high school, college, and rotation PhD students for assistance in collecting egg-laying data: Julia Canick, Emily J Dennis, Solomon Dworkin, Katie Kistler, Stephanie Marcus, Nicholas Urban Schwartz, Russell Shephard, Eva Shrestha, Krithika Venkataraman, and Josh Zeng. We thank Alison Ehrlich and Zachary Gilbert for assistance with molecular biology and mosquito husbandry; Gloria Gordon and Libby Mejia for expert mosquito rearing; Victoria Danan, Nick Didkovsky, Nicolas Renier for contributions to the mosquitobrains.org

project; Raphael Cohn, Gaby Maimon, Vanessa Ruta, John Tuthill, and Ari Zolin for guidance on calcium imaging; James Petrillo, Daniel Gross, and Peer Strogies of the Rockefeller Precision Instrumentation Technology core for design and fabrication of egg-laying assays and a device for calcium imaging; Rob A Harrell II at the Insect Transgenesis Facility at the University of Maryland for CRISPR-Cas9 and transgene injections. We thank Olena Riabinina and Christopher Potter for generously making unpublished QF2/QUAS reagents, data, and protocols available to us prior to publication.

## Additional information

### Funding

| Funder | Author |
| --- | --- |
| Howard Hughes Medical Institute | Benjamin J Matthews Leslie B Vosshall |
| Jane Coffin Childs Memorial Fund for Medical Research | Benjamin J Matthews Meg A Younger |
| Grass Foundation | Meg A Younger |
| Leon Levy Foundation | Meg A Younger |
| Kavli Foundation | Meg A Younger |

The funders had no role in study design, data collection and interpretation, or the decision to submit the work for publication.

### Author contributions

Benjamin J Matthews, Meg A Younger, Conceptualization, Data curation, Formal analysis, Supervision, Funding acquisition, Validation, Investigation, Visualization, Methodology, Writing—original draft, Writing—review and editing; Leslie B Vosshall, Supervision, Funding acquisition, Project administration, Writing—review and editing

### Author ORCIDs

Benjamin J Matthews https://orcid.org/0000-0002-8697-699X
Meg A Younger https://orcid.org/0000-0003-4967-7939
Leslie B Vosshall https://orcid.org/0000-0002-6060-8099

### Ethics

Human subjects: For some experiments, animals were blood-fed on live human subjects. Adult females of all genotypes were blood-fed on a single human subject during mutant generation, for behavioral assays, and calcium imaging. Blood-feeding procedures with live hosts were approved and monitored by The Rockefeller University Institutional Review Board, (IRB protocol LV-0652). Human subjects gave their written informed consent to participate.
Animal experimentation: For routine strain maintenance, animals were blood-fed on live mice . Blood-feeding procedures with live hosts were approved and monitored by The Rockefeller University Institutional Animal Care and Use Committee (IACUC protocol 15772).

### Decision letter and Author response

Decision letter https://doi.org/10.7554/eLife.43963.019
Author response https://doi.org/10.7554/eLife.43963.020

## Additional files

### Supplementary files

• Supplementary file 1. Raw data associated with *Figures 1–3* and *Figures 5–7*.
DOI: https://doi.org/10.7554/eLife.43963.012
• Transparent reporting form

DOI: https://doi.org/10.7554/eLife.43963.013

## Data availability

Source data for all plots in Figures 1-3 and Figure 5-7 are provided in Supplementary file 1.

The following previously published datasets were used:

| Author(s) | Year | Dataset title | Dataset URL | Database and Identifier |
|---|---|---|---|---|
| Matthews B J, McBride CS, De-Gennaro M, Despo O, Vosshall LB | 2016 | The neurotranscriptome of the Aedes aegypti mosquito. | https://www.ncbi.nlm.nih.gov/bioproject/?term=PRJNA236239 | NCBI BioProject, PRJNA236239 |
| Matthews BJ, Dudchenko O, Kingan SB, Koren S, Antoshechkin I, Crawford JE, Glassford JG, Herre M, Redmond SN, Rose NH, Weedall GD, Wu Y, Batra SS, Brito-Sierra CA, Buckingham SD, Campbell CL, Chan S, Cox E, Evans BR, Fansiri T, Filipović I, Fontaine A, Gloria-Soria A, Hall R, Joardar VS, Jones AK, Kay RGG, Kodali VK, Lee J, Lycett GJ, Mitchell SN, Muehling J, Murphy MR, Omer AD, Partidge FA, Peluso P, Aiden AP, Ramasay V, Rašić G, Roy S, Saavedra-Rodriguez K, Sharan S, Sharma A, Laird Smith M, Turner J, Weakley AM, Zhao Z, Akbari OS, Black IV WC, Cao H, Darby AC, Hill CA, Johnston JS, Murphy TD, Raikhel AS, Sattelle DB, Sharakhov IV, White BJ, Zhao L, Aiden EL, Mann RS, Lambrechts L, Powell JR, Sharakhova MV, Tu Z, Robertson HM, McBride CS, Hastie AR, Korlach J, Neafsey DE, Phillippy AM, Vosshall LB | 2018 | Improved reference genome of Aedes aegypti informs arbovirus vector control. | https://www.ncbi.nlm.nih.gov/assembly/GCF_002204515.2/ | NCBI Assembly, GCF_002204515.2 |

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
