## [Decision Letter]

Thank you for submitting your article "The ion channel *ppk301* controls freshwater egg-laying in the mosquito *Aedes aegypti*" for consideration by *eLife*. Your article has been reviewed by three peer reviewers, and the evaluation has been overseen by Kristin Scott as Reviewing Editor and Catherine Dulac as the Senior Editor. The following individual involved in the review of your submission has agreed to reveal their identity: Daniel Tracey (Reviewer #2).

The reviewers have discussed the reviews with one another and the Reviewing Editor has drafted this decision to help you prepare a revised submission.

Summary:

This is a fascinating study that examines the molecular and cellular control of egg laying in the mosquito *Aedes aegypti*. The study is a technical tour-de-force, developing new behavioral assays for egg laying preferences in water vs salt water in *Aedes aegypti* mosquitos, generating new genetic tools for the study of 4 ppk channels, and establishing an imaging preparation for tarsal taste receptor neurons. The basic finding is that neurons that express *ppk301* are located in gustatory sensilla and that female mosquitos with the *ppk301* mutation show reduced in egg laying near water and a tendency to lay more eggs in salt. A model is presented where *ppk301* neurons gate egg laying by feeding into command circuits that both increase egg laying in the presence of water and decrease egg laying in the presence of high salt. Overall, this work represents a strong conceptual and technical achievement, and is certainly worthy of publication in *eLife* once appropriate revisions are made.

Essential revisions:

1) A more in-depth characterization of *ppk301* neuron tuning, using calcium imaging, is necessary to address whether the "salt" response is specific to salt, or a general response to solutes. The minimum number of stimuli to determine whether these neurons respond to changes in osmolarity or specifically to salt and water should be examined.

The crux of the problem is that, at the moment, there is no stimulus demonstrated to give a smaller response than water. Thus, we don't know what property of the water *ppk301* neurons are responding to. The implication is that it is the low osmolality, but this is not explicitly demonstrated. It is formally possible that these neurons respond to all liquids, and that water actually gives the smallest response of any liquid stimulus. A broader examination of the receptive fields of these neurons is necessary to resolve this issue.

2) Calcium imaging of *ppk301* neurons in *ppk301* mutants in response to salt and water, which would presumably show a change in the water response compared to controls. This would greatly assist in linking the mutant behavior with the physiology of the neurons.

Other points:

1) This study clearly demonstrated that *ppk301* is involved in egg-laying behavior in females. But it is unclear if *ppk301*-expressing neurons have instructive or permissive signals for this behavior. The authors favor an instructive model where the activation of *ppk301* is directly linked to egg-laying. However, based on the functional data (Figure 5), *ppk301*-expressing neurons may simply transmit liquid information (e.g., existence of liquid) regardless of salinity, which provides permissive signals for egg-laying behavior. Subsequently, other pathways may mediate instructive signals for laying or not laying eggs based on salt concentration. These two models are testable by gain-of-function manipulation on *ppk301* neurons: if the authors' model is correct, artificial activation of these neurons would drive egg-laying behavior without proximal water. Nevertheless, these experiments require a generation of new reporter line. If this is not feasible, it is worth considering a permissive model in Figure 5G.

2) To more easily compare the results with artificial seawater to those with NaCl, it would be nice if the salt concentration were given for each percentage of artificial seawater.

3) The authors should cite:

Gustatory receptor neuron responds to DEET and other insect repellents in the yellow-fever mosquito, Aedes aegypti

By: Sanford, Jillian L.; Shields, Vonnie D. C.; Dickens, Joseph C.

NATURWISSENSCHAFTEN Volume: 100 Issue: 3 Pages: 269-273 Published: MAR 2013

which has shown the existence of salt sensing neurons in *Aedes* (that don't also respond to water).

---

## [Author Response]

Essential revisions:1) A more in-depth characterization of ppk301 neuron tuning, using calcium imaging, is necessary to address whether the "salt" response is specific to salt, or a general response to solutes. The minimum number of stimuli to determine whether these neurons respond to changes in osmolarity or specifically to salt and water should be examined.The crux of the problem is that, at the moment, there is no stimulus demonstrated to give a smaller response than water. Thus, we don't know what property of the water ppk301 neurons are responding to. The implication is that it is the low osmolality, but this is not explicitly demonstrated. It is formally possible that these neurons respond to all liquids, and that water actually gives the smallest response of any liquid stimulus. A broader examination of the receptive fields of these neurons is necessary to resolve this issue.

This is a valid and important concern. We have carried out new imaging experiments with two additional non-ionic solutes: L-serine and D-(+)-cellobiose at 200 mOsm/kg. This osmolality gives the greatest response to NaCl and is a concentration of NaCl that is very aversive in the context of the egg-laying behavior that we are studying. The results indicate that neither L-serine nor D-(+)-cellobiose activated *ppk301* neurons to the level seen with NaCl. The response to D-(+)-cellobiose was comparable to water and the response to L-serine was only slightly reduced in comparison to water.

To accompany these experiments, we also conducted new egg-laying behavior experiments to determine the preference for 200mOsm/kg L-serine and D-(+)-cellobiose in comparison to water. We saw that animals prefer to lay on freshwater when presented with 200 mOsm/kg NaCl, as we had previously reported. However, animals displayed no preference for water over either 200 mOsm/kg L-serine or D-(+)-cellobiose, indicating that the behavior is salt-specific and not driven by osmolality at this osmotic pressure.

While we cannot rule out that ppk301-expressing cells are inhibited by very high dissolved solute concentrations as is seen in *D. melanogaster* (>400mOsm), it is far outside the range of the behavior that we tested and cannot account for the egg laying preference against NaCl.

2) Calcium imaging of ppk301 neurons in ppk301 mutants in response to salt and water, which would presumably show a change in the water response compared to controls. This would greatly assist in linking the mutant behavior with the physiology of the neurons.

The paper was delayed in resubmission for these many months so that we could comply with this request, which required significant genetic crossing to generate the requested strain. We are pleased that we ultimately succeeded in building the strain required to image activity of neurons labeled in the *ppk301*-QF2 strain in the context of a homozygous *ppk301* mutant background. The new results show that the response to freshwater is almost entirely abolished in the *ppk301* mutant. While there is some reduction in the response to NaCl, the overall salt response remains intact in the *ppk301* mutant animals. These data are consistent with the model that ppk301 is necessary for the water component of the response and that there is a non-*ppk301* dependent salt response in these cells. The identity of this non-*ppk301* salt receptor remains to be elucidated. We expand upon these findings in detail in a revised discussion with a new model that incorporates our additional experimental data.

We thank the reviewers for these suggestions and feel that in combination these new experiments substantially strengthen the original conclusions of the paper.

Other points:

*1) This study clearly demonstrated that ppk301 is involved in egg-laying behavior in females. But it is unclear if ppk301-expressing neurons have instructive or permissive signals for this behavior. The authors favor an instructive model where the activation of ppk301 is directly linked to egg-laying. However, based on the functional data (Figure 5), ppk301-expressing neurons may simply transmit liquid information (e.g., existence of liquid) regardless of salinity, which provides permissive signals for egg-laying behavior. Subsequently, other pathways may mediate instructive signals for laying or not laying eggs based on salt concentration. These two models are testable by gain-of-function manipulation on ppk301 neurons: if the authors' model is correct, artificial activation of these neurons would drive egg-laying behavior without proximal water. Nevertheless, these experiments require a generation of new reporter line. If this is not feasible, it is worth considering a permissive model in Figure 5G.*

We agree with the reviewer that exogenous opto/thermo/chemogenetic activation of these neurons would be a great way to test whether *ppk301* neuron activation directly instructs egg-laying or merely acts as a permissive signal to enable egg-laying behavior. The reviewer is also correct that carrying out this experiment would require substantial effort in generating, testing, and validating new transgenic mosquito strains. We are at the early stages of generating optogenetic tools in the mosquito, but these are not at the point that we would feel confident interpreting the results. In lieu of carrying out the requested experiments, we have edited the model and discussion in Figure 5G as suggested by the reviewer.

*2) To more easily compare the results with artificial seawater to those with NaCl, it would be nice if the salt concentration were given for each percentage of artificial seawater*.

We have generated a supplementary table with the molarity of each ion in the artificial seawater solutions and the NaCl solutions used in behavior. There are 11 distinct ions in the artificial seawater solution derived from 10 salts, and we have included each individual ion and the molarity of that ion in each dilution of artificial seawater, since this is the property that is most likely to affect the behavior of the mosquito. We do not expect that the behavior will track perfectly with Na^+^ or Cl^-^ concentration because artificial seawater contains many other salts that are known to affect physiology. However, a direct comparison can be made from the numbers listed in the table. The entire description of its composition is also in the cited reference (Kester et al., 1967). The complexity of artificial seawater is what prompted us to switch to NaCl for all future experiments.

It should be noted that for all other behavior and imaging experiments, solutions were made and the osmolality was empirically measured and adjusted using an osmometer. The preference for accuracy in osmolality rather than expected molarity based on weight means that the molarities listed for the NaCl solutions are the expected values for these solutions and not the measured values. We chose not to make any theoretical corrections based on environmental conditions, as we did not measure these at the time of our experiments.

3) The authors should cite:Gustatory receptor neuron responds to DEET and other insect repellents in the yellow-fever mosquito, Aedes aegyptiBy: Sanford, Jillian L.; Shields, Vonnie D. C.; Dickens, Joseph C.NATURWISSENSCHAFTEN Volume: 100 Issue: 3 Pages: 269-273 Published: MAR 2013which has shown the existence of salt sensing neurons in Aedes (that don't also respond to water).

The citation has been added.